# DiffDomain enables identification of structurally reorganized topologically associating domains

Dunming Hua[1,2,7], Ming Gu[1,2,7], Xiao Zhang [1,2,7], Yanyi Du[1,2,7], Hangcheng Xie[1,2], Li Qi[3], Xiangjun Du [1,2], Zhidong Bai [4], Xiaopeng Zhu [5,6] & Dechao Tian [1,2] ✉

Topologically associating domains (TADs) are critical structural units in three-dimensional genome organization of mammalian genome. Dynamic reorganizations of TADs between health and disease states are associated with essential genome functions. However, computational methods for identifying reorganized TADs are still in the early stages of development. Here, we present DiffDomain, an algorithm leveraging high-dimensional random matrix theory to identify structurally reorganized TADs using high-throughput chromosome conformation capture (Hi−C) contact maps. Method comparison using multiple real Hi−C datasets reveals that DiffDomain outperforms alternative methods for false positive rates, true positive rates, and identifying a new subtype of reorganized TADs. Applying DiffDomain to Hi−C data from different cell types and disease states demonstrates its biological relevance. Identified reorganized TADs are associated with structural variations and epigenomic changes such as changes in CTCF binding sites. By applying to a single-cell Hi−C data from mouse neuronal development, DiffDomain can identify reorganized TADs between cell types with reasonable reproducibility using pseudo-bulk Hi−C data from as few as 100 cells per condition. Moreover, DiffDomain reveals differential cell-to-population variability and heterogeneous cell-to-cell variability in TADs. Therefore, DiffDomain is a statistically sound method for better comparative analysis of TADs using both Hi−C and single-cell Hi−C data.

The recent development of mapping technologies such as Hi−C[1] that probes the 3D genome organization reveals that a chromosome is divided into topologically associating domains (TADs)[2,3]. TADs are genomic regions where chromatin loci more frequently interact with other chromatin loci within the TAD than with those from outside of the TAD. TADs are functional units for transcriptional regulation by constraining interactions between enhancers and promoters[4], for example. Although TADs are stable between cell types as revealed by earlier studies[2,5], there is growing evidence for TAD reorganization in diseases[6–8], cell differentiation[5,9,10], somatic cellular reprogramming[11], between neuronal cell types[12], and between species[13,14]. For example, extensive reorganizations of TADs are observed during somatic cell

---

[1]Department of Biostatistics and Systems Biology, School of Public Health (Shenzhen), Sun Yat-sen University, Shenzhen, Guangdong 510275, China. [2]Department of Biostatistics and Systems Biology, School of Public Health (Shenzhen), Shenzhen Campus of Sun Yat-sen University, Shenzhen 518107, China. [3]Chongqing Municipal Center for Disease Control and Prevention, Chongqing 400042, China. [4]KLASMOE & School of Mathematics and Statistics, Northeast Normal University, Changchun, Jilin 130024, China. [5]MyCellome LLC., Allison Park, PA 15101, USA. [6]Computational Biology Department, School of Computer Science, Carnegie Mellon University, Pittsburgh, PA 15213, USA. [7]These authors contributed equally: Dunming Hua, Ming Gu, Xiao Zhang, Yanyi Du. ✉e-mail: tiandch@mail.sysu.edu.cn

reprogramming, associating with dynamics of transcriptional regulation and changes in cellular identity[11]. TADs are also variable among individual cells, as revealed by single-cell studies[15-20] and live-cell imaging[21]. Thus, it is important to identify reorganized TADs through comparative analysis to further understand the functional relevance of 3D genome organization, a major priority of current work in the field[22].

The majority of current methods call a reorganized TAD if at least one of its two boundaries changed between two conditions[11,12,23-28]. These methods enable easy integration with other analysis pipelines and identify reorganized TADs with clear biological interpretation. However, they fail to identify reorganized TADs without changes in boundaries, in addition to lacking statistical tests to differentiate random perturbations and significant structural reorganization of a TAD. Only a few nonparametric statistical methods are proposed to call TAD reorganization[29-32]. These methods define the structural similarity of a TAD by statistics from two Hi-C contact matrices, such as the stratum-adjusted correlation coefficient used by DiffGR[30]. Distributions of the statistics on pairs of simulated Hi-C matrices are then used to compute empirical $P$ values. However, these nonparametric statistical methods are conservative (see our own comparison later). TADs in high-resolution Hi-C data are relatively small. The median size of TADs is 185 kb[33]. The small size feature of TADs poses another computational challenge for identifying structurally rewired TADs using low-resolution Hi-C data. Importantly, identifying reorganized TADs using emerging single-cell Hi-C (scHi-C) data is largely underexplored. Other methods are developed for comparing Hi-C matrices at different scales and for different purposes: quantifying similarities of genome-wide Hi-C contact matrices[34,35], identifying differential A/B compartments[36], and identifying differential chromatin interactions[37-39]. However, these methods are not tailored to compare Hi-C contact matrices at the TAD level, which is not optimal for identifying reorganized TADs (see our own comparison later). Therefore, new algorithms are needed to fill these gaps.

Here, we develop DiffDomain, a new parametric statistical method for identifying reorganized TADs. Its inputs are two Hi-C contact matrices from two biological conditions and a set of TADs called in biological condition 1. This setting enables straightforward integration of DiffDomain with other analysis pipelines of Hi-C data, such as TAD calling and integrative analysis of multi-omics data. For each TAD, DiffDomain directly computes a difference matrix and then normalizes it properly, skipping the challenging normalization steps for individual Hi-C contact matrices. DiffDomain then borrows well-established theoretical results in random matrix theory to compute a $P$ value. We show that the assumptions of DiffDomain are reasonable. Method comparisons on real data reveal that DiffDomain has substantial advantages over alternative methods in false positive rates and accuracy in identifying truly reorganized TADs. Reorganized TADs identified by DiffDomain are biologically relevant in different human cell lines and disease states. Application to scHi-C data reveals that DiffDomain can identify reorganized TADs between cell types and TADs with differential variabilities among individual cells within the same cell type. Moreover, DiffDomain can quantify cell-to-cell variability of TADs between individual cells. Together, these analyses demonstrate the power of DiffDomoain for better identification of structurally reorganized TADs using both bulk Hi-C and single-cell Hi-C data.

## Results
### Overview of DiffDomain
The workflow of DiffDomain is illustrated in Fig. 1. Its input is a set of TADs called in biological condition 1 and their corresponding Hi-C contact matrices from biological conditions 1 and 2 (Fig. 1a). In this paper, TADs are called by Arrowhead[33] and Hi-C contact matrices are KR-normalized, unless otherwise stated (Supplementary Method 1). Our goal is to test if each TAD identified in biological condition 1 has

significant structural reorganization in biological condition 2. The core of DiffDomain is formulating the problem as a hypothesis testing problem, where the null hypothesis is that the TAD doesn't undergo significant structural reorganization in condition 2. To achieve this goal, for each TAD with $N$ bins, DiffDomain extracts the $N \times N$ KR-normalized Hi-C contact matrices specific to the TAD region from the two biological conditions, which are denoted as $A_1$ and $A_2$ (Fig. 1a). Note that $A_1$ and $A_2$ are $N \times N$ submatrices of the genome-wide Hi-C contact matrices. DiffDomain first log-transform them to adjust for the exponential decay of Hi-C contacts with increased 1D distances between chromosome bins. Their difference $\log(A_1) - \log(A_2)$ is calculated and denoted by $D$ (Fig. 1b). $D$ is further normalized by a 1D distance-stratified standardization procedure, similar to the procedures in HiC-DC+[38] and SnapHiC[40]. Specifically, each $d$-off diagonal part of $D$ is subtracted by its sample mean and divided by its sample standard deviation (Fig. 1c), $-N + 2 \leq d \leq N - 2$, reducing 1D distance-dependence of values in $D$ and differences caused by variation in read depths between two biological conditions (see Supplementary Fig. 1 for two more detailed visualization). Intuitively, if a TAD is not significantly reorganized, normalized $D$ would resemble a white noise random matrix, enabling us to borrow theoretical results in random matrix theory. Under the null hypothesis, DiffDomain assumes that $D/\sqrt{N}$ is a generalized Wigner matrix (Fig. 1d), a well-studied random matrix model. Its largest eigenvalue $\lambda_N$ is proved to be fluctuating around 2. Armed with this fact, DiffDomain reformulates the reorganized TAD identification problem into the hypothesis testing problem:

$$H_0 : \lambda_N = 2 \text{ vs. } H_1 : \lambda_N > 2. \tag{1}$$

The key theoretical results empowering DiffDomain is that $\theta_N = N^{2/3}(\lambda_N - 2)$, a normalized $\lambda_N$, asymptotically follows a Tracy-Widom distribution with $\beta = 1$, denoted as $TW_1$. Thus, $\theta_N$ is chosen as the test statistic, and the one-sided $P$ value is calculated as $P_{TW_1}(\theta_N \geq x)$. $H_0$ is rejected if the $P$ value is less than a predefined significant level $\alpha$, which is 0.05 in this paper (Fig. 1e). The pseudocode is shown in Supplementary Method 2. For a set of TADs, $P$ values are adjusted for multiple comparisons using the Benjamini-Hochberg (BH) method as the default. Once DiffDomain identifies the subset of reorganized TADs, it further classifies them into six subtypes based on changes in their boundaries, which is beneficial for downstream biological analyses and interpretations (Fig. 1f, Supplementary Method 3). TAD reorganization subtypes are verified by aggregation peak analyses (APA) on multiple real datasets (Supplementary Fig. 2). A few reorganized TADs in real Hi-C data are shown in Fig. 1g. Details are described in the Methods section.

Note that although each biological condition may have multiple Hi-C replicates, DiffDomain takes the combined Hi-C contact matrix from the replicates as the input, which is a common practice to generate a large number of Hi-C interactions[33]. Correlations among Hi-C interactions lead to correlations among entries in the $D/\sqrt{N}$ matrix, violating the independent assumption among upper diagonal entries of generalized Wigner matrix. However, the violation of the independence assumption does not substantially alter the properties of $D/\sqrt{N}$ and DiffDomain based on empirical analysis results, suggesting that assumptions of DiffDomain are appropriate (see Methods section, Supplementary Note 1, Supplementary Fig. 3). When $N = 10$, the Tracy-Widom distribution $TW_1$ is an adequate approximation of the exact distribution of $\theta_N$[41]. Under common 10 kb resolution Hi-C data, $N = 10$ refers to TADs with 100 kb in length, much smaller than the median TAD length of 185 kb[33]. Thus DiffDomain only computes the $P$ value for TADs with at least 10 chromosome bins, a practical constraint. DiffDomain is robust to a varied number of sequencing reads, Hi-C resolution, and different TAD callers (Supplementary Note 2, Supplementary Figs. 4 and 5).

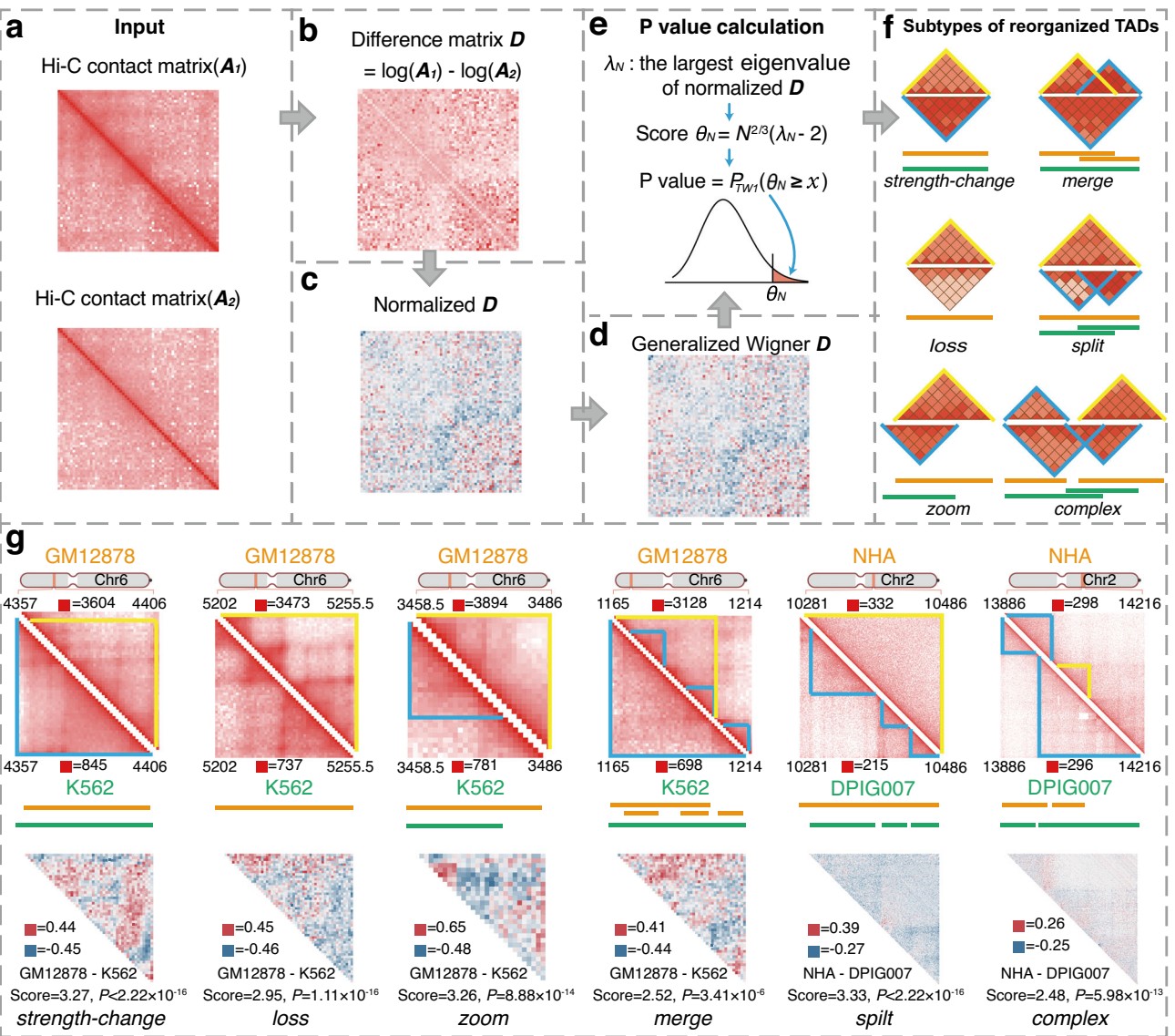

**Fig. 1 | DiffDomain workflow and example outputs. a** Input are a TAD in condition 1 and its two Hi-C contact matrices ($A_1$ and $A_2$) in two biological conditions 1 and 2. **b** Difference between log-transformed $A_1$ and $A_2$, which is denoted as $D$. **c** Normalization of $D$ by a 1D distance-stratified standardization procedure. Its $d$-off diagonal part is normalized by $d$-off diagonal part-specific sample mean and sample standard deviation. **d** $D$ is transformed by dividing $\sqrt{N}$. Under the null hypothesis, it is assumed to be a generalized Wigner matrix. **e** One-sided $P$ value is calculated based on the fact that $\theta_N$, normalized largest eigenvalue of $D$, follows Tracy-Widom distribution with $\beta = 1$ (denoted as $TW_1$ distribution). A TAD is identified as a reorganized TAD if $P$ value $\leq 0.05$. **f** Reorganized TADs are classified into six subtypes based on changes in TAD boundaries. The heatmap diagram illustrates TADs in

condition 1 (*Top*) and condition 2 (*Bottom*), with lines representing TAD regions of the same condition sharing the same color. **g** Example of the subtypes of reorganized TADs. Data are from two studies[33,74]. *Top:* upper and lower triangular matrices represent Hi-C data in conditions 1 and 2, with blue triangles representing TADs and yellow triangles representing reorganized TADs; *Middle:* TAD regions from the same condition are represented by lines of the same color; *Bottom:* upper triangular section of the normalized difference matrix $D/\sqrt{N}$ computated from the two Hi-C matrices in the *Top* section. Red and blue boxes represent the maximum and minimum values in the visualized matrices. The score and one-sided $P$ value are computed by DiffDomain. $P$ values smaller than $2.22 \times 10^{-16}$ are denoted by $P < 2.22 \times 10^{-16}$.

## DiffDomain consistently outperforms alternative methods in multiple aspects

First, we assess the false positive rate (FPR) which is the ratio of the number of false positives to the number of true negatives. A smaller FPR means that the identified significantly reorganized TADs are more likely to be true. Due to the lack of gold-standard data, we resort to analyzing the proportions of significantly reorganized TADs between five different Hi-C replicates from the GM12878 cell line. These Hi-C replicates are generated by different experimental procedures and have a highly varied total number of Hi-C contacts (Supplementary Table 1). However, the TADs are expected to have few structural changes between these Hi-C replicates. The proportion of identified

reorganized TADs is treated as an estimate of FPR (Supplementary Method 4) and is expected to be small. The Hi-C resolution is chosen as 10 kb. Comparing GM12878 Hi-C replicates *primary* and *replicate*, we find that DiffDomain, TADCompare[31], and HiCcompare[42] have FPRs that are close to the given significant level of 0.05, suggesting good controls of FPR. In contrast, DiffGR[30], DiffTAD[29], and HiC−DC+[38] have inflated FPRs (more than two-fold higher than 0.05), indicating poor controls of FPR (Fig. 2a). Similar results are observed by repeating the above analysis to other GM12878 Hi-C replicates and 25 kb resolution Hi-C data (Supplementary Fig. 6).

Good control in FPR does not necessarily represent high power in detecting reorganized TADs between biological conditions.

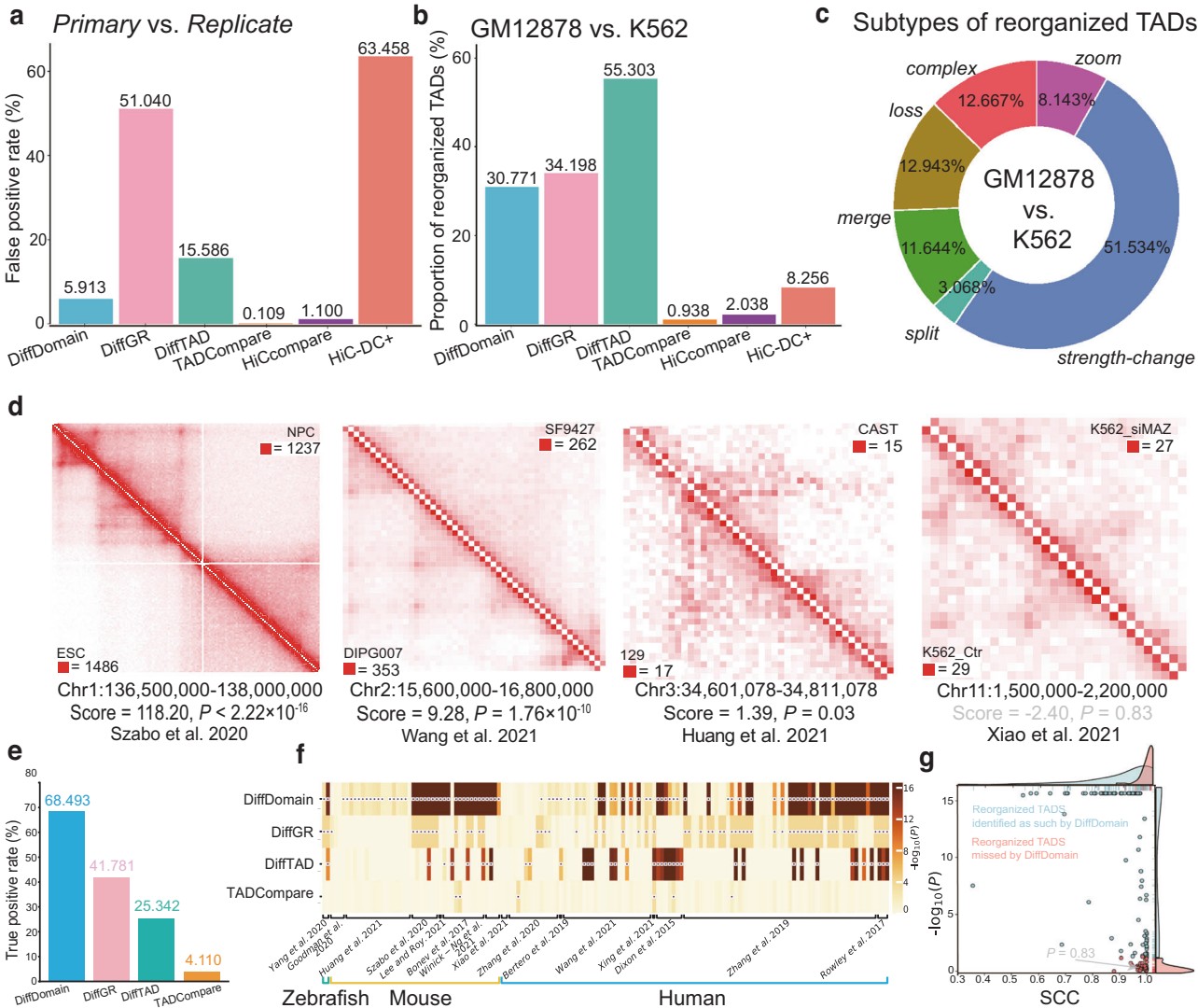

**Fig. 2 | Benchmarking DiffDomain against alternative methods. a** FPRs of Diff-Domain and alternative methods in comparing two GM12878 Hi-C replicates (*primary* and *replicate*). FPRs in comparing other Hi-C replicates of GM12878 are shown in Supplementary Fig. 6. FPR equals the ratio of the number of identified reorganized TADs to the number of TADs in GM12878. **b** Proportions of identified reorganized TADs by DiffDomain and alternative methods when comparing blood-related cell lines GM12878 and K562. TADs are GM12878 TADs. Results on other pairs of human cell lines are reported in Supplementary Fig. 7. **c** Percentages of the subtypes of reorganized TADs when comparing GM12878 and K562. Results on other pairs of human cell lines are reported in Supplementary Fig. 9. **d** Heatmaps showing Hi-C contact matrices of four truly reorganized TADs. Upper and lower triangular matrices represent Hi-C data in conditions 1 and 2. The scores and unadjusted one-sided *P* value below the heatmap are computed by DiffDomain. Among them, three are correctly identified as such by DiffDomain (unadjusted

$P \le 0.05$, true positives), and one is missed by DiffDomain (unadjusted $P > 0.05$, false negative). Truly reorganized TADs are manually collected and treated as the gold standard positives (see Supplementary Method 5 for more details). **e** Barplot showing TPRs of DiffDomain and alternative methods. TPR equals the ratio of the number of reorganized TADs that are identified as such (true positives) to the number of reorganized TADs (positives). **f** Heatmap showing unadjusted one-sided *P* values computed by DiffDomain for testing truly reorganized TADs. The purple dot with a white border represents $P < 0.05$. **g** Scatter points of unadjusted one-sided *P* values by DiffDomain (*y*-axis) and SCCs[34] (*x*-axis) when testing the truly reorganized TADs. Detailed information, including reference, TAD region, data accession number, and species for the truly reorganized TADs (Fig. 2e, f), is presented in Supplementary Method 5 and Supplementary Table 3. *P* values smaller than $2.22 \times 10^{-16}$ are denoted by $P < 2.22 \times 10^{-16}$. Abbreviation: SCC stratum-adjusted correlation coefficient.

To investigate this, we compare TADs between two blood cell lines, GM12878 and K562. DiffDomain identifies that 30.771% of GM12878 TADs are reorganized in K562. In contrast, TADCompare, HiCcompare, and HiC-DC+ only identify ≤8.256% of GM12878 TADs that are reorganized in K562 (Fig 2b), suggesting that they are too conservative in identifying reorganized TADs between biological conditions. Similar results are observed by repeating the above analysis to other human cell lines and 25 kb resolution Hi-C data (Supplementary Table 2, Supplementary Fig. 7), demonstrating the robustness of the observations. Conservation of TADCompare is because it is designed to scan every chromatin loci for potential reorganized TAD boundaries.

But this analysis uses a given list of TADs, a common practice in Hi-C data analysis, which sharply narrows down the search space of TAD-Compare. Conservations of HiCcompare and HiC-DC+ are because they are designed for detecting differential chromatin interactions, not specifically tailored for identifying reorganized TADs.

Compared with TADsplimer which specifically identifies *split* and *merge* TADs[25], DiffDomain identifies similar numbers of *split* and *merge* TADs between multiple pairs of human cell lines (Supplementary Fig. 8). Importantly, DiffDomain identifies that the majority (minimum 43.137%, median 81.357%, maximum 98.022%) of the identified reorganized TADs are the other four subtypes (Supplementary Fig. 9),

which can not be detected by TADsplimer. For example, among the GM12878 TADs that are identified as reorganized in K562 by DiffDomain, *strength-change* is the leading subtype of reorganized TADs, consistent with the fact that both GM12878 and K562 are blood cell lines (Fig 2c). These results demonstrate that DiffDomain has substantial improvements over TADsplimer.

We next investigate the true positive rate (TPR). A higher TPR means that more truly reorganized TADs are correctly identified as reorganized TADs. Through an extensive literature search, we collect 65 TADs that are reorganized between 146 pairs of biological conditions in 15 published papers (Supplementary Table 3, Supplementary Method 5). We use these TADs as the gold standard data to compute the TPR (Supplementary Method 4) and also call these TADs truly reorganized TADs. Four truly reorganized TADs, either correctly identified or missed by DiffDomain, are shown in Fig. 2d. HiCcompare and HiC−DC+, designed for identifying differential chromatin interactions, are not directly applicable to the only testing reorganization of one single TAD and thus are excluded from the analysis. We find that the TPR of Diff-Domain is 68.493%, which is 1.639, 2.703, and 16.665 times higher than that of alternative methods DiffGR, DiffTAD, and TADCompare, respectively (Fig. 2e). Compared with DiffDomain, DiffGR, DiffTAD, and TADCompare only uniquely identify 11, 10, and 1 truly reorganized TADs, respectively (Supplementary Fig. 10). Closer examination shows that DiffDomain has much smaller $P$ values than other methods (Wilcoxon rank-sum test, $P \leq 2.31 \times 10^{-8}$, Fig. 2f), demonstrating that DiffDomain has stronger statistical evidence in favor of truly reorganized TADs. Based on the depictions of TAD changes reported in the publications, the truly reorganized TADs are broadly categorized into three groups: domain-level change, boundary-level change, and loop-level change (Supplementary Method 5). These groups have decreased reorganization levels with increased stratum-adjusted correlation coefficient (SCC) scores[34] between biological conditions (Supplementary Fig. 11a). Across the groups, DiffDomain consistently achieves the highest TPRs, while the second-best method varies (Supplementary Fig. 11b), further demonstrating the advantages of DiffDomain over alternative methods. DiffDomain still misses 31.507% of possible pairwise comparisons of truly reorganized TADs. One reason is that some of the missed truly reorganized TADs have highly similar Hi−C contact matrices between biological conditions. For example, the missed truly reorganized TAD, chr11:1500000−2200000 (Fig. 2d), has the SCC score at 0.998. Generally, missed truly reorganized TADs have significantly ($P = 8.26 \times 10^{-6}$) higher SCC scores than those correctly identified reorganized TADs by DiffDomain (Fig. 2g). Similar results are observed when stratifying by the groups of truly reorganized TADs (Supplementary Fig. 11c). Because DiffGR uses SCC as the test statistic, these results also partially explain the low TPR (41.781%) of DiffGR and highlight that SCC alone is not optimal for identifying reorganized TADs. Another reason is that the resolutions of some Hi−C data are low since $P$ values are moderately negatively associated with the maximum values of Hi−C contact matrices (Spearman's rank correlation coefficient $\rho = -0.534$).

Additionally, DiffDomain is efficient in memory usage and acceptable in computation time compared with alternative methods (Supplementary Note 3, Supplementary Fig. 12).

In summary, compared with alternative methods, DiffDomain has multiple improvements, including FPRs, proportions of identified reorganized TADs between different biological conditions, subtypes of reorganized TADs, and TPRs.

## Reorganized TADs are associated with epigenomic changes

Armed with the advantages of DiffDomain over alternative methods, we explore the connections between TAD reorganization and epigenomic dynamics. We first showcase a GM12878 TAD that is significantly reorganized in K562 and classified as a *strength-change* TAD by DiffDomain (Fig. 3). The TAD covers a 445 kb region on chromosome 6. The TAD structural changes involve the vascular endothelial

growth factor gene *VEGFA*, which is a major tumor angiogenic gene that is over-expressed in leukemia (see reviews[43,44] for more details), consistent with the fact that K562 cells are chronic myelogenous leukemia cells. We find that the reorganized TAD has K562-specific functional annotations. The genomic region covered by the TAD is more accessible (1.71 times higher DNase peak coverage) in K562 than in GM12878 (Fig. 3b). The H3K27ac and H3K4me1 peak coverages of the TAD region in K562 are 3.24 and 3.28 times higher than the coverages in GM12878, respectively. In contrast, the H3K4me3 and H3K36me3 peak coverages of the TAD in K562 are only 1.31 and 1.25 times higher than the coverages in GM12878, respectively. Four regions in the TAD are annotated as super-enhancers[45] only in K562 (Fig. 3b). Note that the TAD region is in A compartments in both cell types, suggesting that the A/B compartments switch is not the reason for the gain in accessibility and histone modifications that are associated with gene activation. The normalized difference matrix **D** between Hi−C contact matrices of the TAD highlights that super-enhancer SE2 has increased Hi−C contacts with the *VEGFA* gene in K562 (Fig. 3c). To gain further insights into structural differences of the TAD in the two cell types, we compare the 3D structural representations of the TAD region. We run Chrom3D[46] 100 times to construct 100 possible 3D structures in each cell type for statistical comparisons. Two possible 3D structures with each per cell type illustrate the 3D structural differences of the TAD between GM12878 and K562 (Fig. 3d, e). Overall, the super-enhancer SE2, but not SE3 (Fig. 3b), is much spatially closer ($P < 2.22 \times 10^{-16}$) to VEGFA in K562 than in GM12878 (Supplementary Fig. 13). These results show that the reorganized GM12878 TAD in K562 has K562-specific chromatin organization and potential biological functions.

Generally, comparative analyses across multiple pairs of human normal and disease cell lines reveal that *strength-change* reorganized TADs with increased contact frequencies have significant increase in the number of CTCF binding sites at TAD boundaries compared with other TADs, whereas lost TAD boundaries associated with *loss*, *zoom*, and *merge* TADs have significantly fewer number of CTCF binding sites, consisting with enrichment of CTCF binding sites in TAD boundaries (Supplementary Note 4, Supplementary Fig. 14). Across diverse human cell lines, while TADs remain relatively stable, their proportions of reorganized TADs vary depending on the cell type, and these variations can cluster cell lines with similar cell identities (Supplementary Note 5, Supplementary Figs. 7 and 15). GM12878 (normal lymphoblastoid cell line) TADs that are reorganized in K562 (chronic myeloid leukemia cell lines) are enriched ($P = 0.01$) in disease genes in chronic myelogenous leukemia. Across pairs of cell types, reorganized TADs tend to gain in chromatin accessibility and active transcription signals H3K27ac and K3K4me1. Particularly, TAD reorganization subtypes have distinct associations with chromatin accessibility as well as histone modifications. Specifically, TAD reorganization subtypes *strength-change-up*, *zoom*, *split*, and *complex* are associated with increased chromatin accessibility and histone modification signals marking active transcription activities. Conversely, TAD reorganization subtypes *loss*, *strength-change-down*, and *merge* are associated with decreased histone modifications signals marking active transcription activities, emphasizing the importance of TAD reorganization subtypes in investigating genome activity and functionality (Supplementary Note 6, Supplementary Figs. 16−18). Compared to normal human astrocytes (NHA), patient-derived diffuse intrinsic pontine glioma cell lines DIPG007 and DIPGXIII share a substantial proportion (73.46%) of reorganized TADs, some harboring potential oncogenes and super-enhancers, while dBET6 treatment demonstrates a stronger effect on TAD reorganization than BRD4 inhibition (Supplementary Note 7, Supplementary Figs. 19−21, Supplementary Table 4). Together, these results demonstrate the functional relevance of reorganized TADs in multiple human normal and disease cell lines.

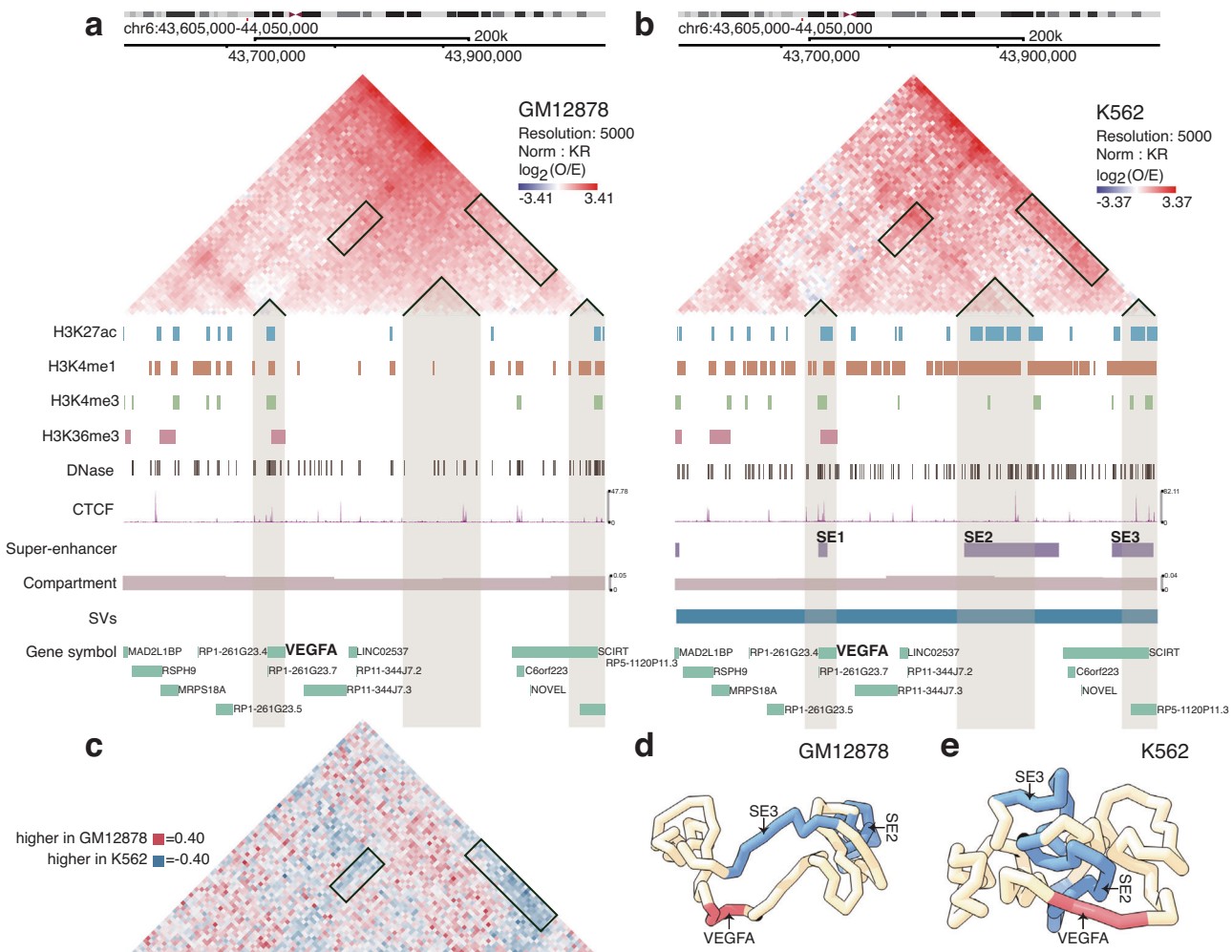

**Fig. 3 | A reorganized TAD involves the angiogenic gene *VEGFA*.** Hi–C contact matrices and 1D genomic tracks of the TAD region (Chr6:43605000-44050000) in GM12878 **a** and K562 **b** cell lines that show differences in both Hi–C data and epigenomics profiles involving *VEGFA* gene. **c** The normalized difference matrix ***D*** between the two cell lines highlights the differences in Hi–C contact maps.

Rectangular boxes highlight the increased Hi–C interactions in K562. Rectangular boxes in panels **a**, **b** highlight corresponding sections in Hi–C contact maps. **d, e** Potential 3D structures of the TAD in GM12878 and K562. They are estimated by Chrom3D[46] and demonstrate the TAD structural differences between GM12878 and K562.

## Reorganized TADs are enriched with structural variations (SVs)

SVs can contribute to diseases by rewiring 3D genome organization. To further demonstrate the biological relevance of reorganized TADs, we systematically investigate the associations between SVs and reorganized TADs. High-resolution SVs, including deletions and duplications, from erythroleukemia (K562 cell line) and pediatric high-grade glioma (DIPG007 and DIPGXIII cell lines) are downloaded from Wang et al.[47]. Because Arrowhead TADs does not necessarily cover the whole genome, SVs are filtered by keeping only those with their genomic regions overlapping with TADs (illustrated by two examples in Fig. 4a). The number of SVs and paired normal Hi–C data are summarized in Supplementary Table 5.

If an SV region overlaps with one reorganized TAD, we consider the SV to have an associated reorganized TADs. We find that the majority (72.2%) of the SVs have such associations (Fig 4b), with proportions significantly higher than those of randomly sampled, equal-numbered reorganized TADs (Supplementary Fig. 22). SVs are associated with distinct abnormal patterns in Hi–C contact maps and are categorized into four types: deletions and duplications with 5' to 3' fusion, 5' to 5' fusion, and 3' to 3' fusion[47]. When stratified by SV types, the majority (>55%) of SVs with the same type also have associated reorganized TADs (Fig 4b). However, each type of SVs has distinct associations with the

subtypes of reorganized TADs. For example, in the comparison between GM12878 and K562 cell lines, the reorganized TADs associated with the four types of K562 SVs have differential distributions over their subtypes (Fig. 4c). The leading subtype of reorganized TADs, *strength-change*, is consistently observed across the four types of SVs. However, the second leading subtype of reorganized TADs varies among the four types of SVs (Fig. 4c). This observation is further emphasized by evident differences in the APA plots (Fig. 4d). Importantly, the association between SV type and TAD reorganization subtype is disease-specific, supported by the clear distinctions in both the APA plots and the subtype distributions of reorganized TADs across K562, DIPG007, and DIPGXIII cell lines (Fig. 4c, d). Particularly, leading reorganized TAD subtypes associated with SVs vary among cell types, with *strength-change* and *loss* in K562; *strength-change*, *zoom*, and *split* in DIPG007; and *loss* in DIPGXIII (Fig. 4c). This variability may be due to the substantial differences in SV lengths among these cell types (Fig. 4e). Notably, small proportions (17.4–27.7%, 8 in K562, 10 in DIPG007, and 4 in DIPGXIII) of SVs lack associated reorganized TADs (Fig. 4b). Upon visual examination through the Nucleome Browser, in total, 13 SVs have reorganized TADs that are not detected by DiffDomain, implying that the remaining 9 SVs in the three cell types may lack associated TAD reorganization (Supplementary Figs. 23–25). Nevertheless, these results

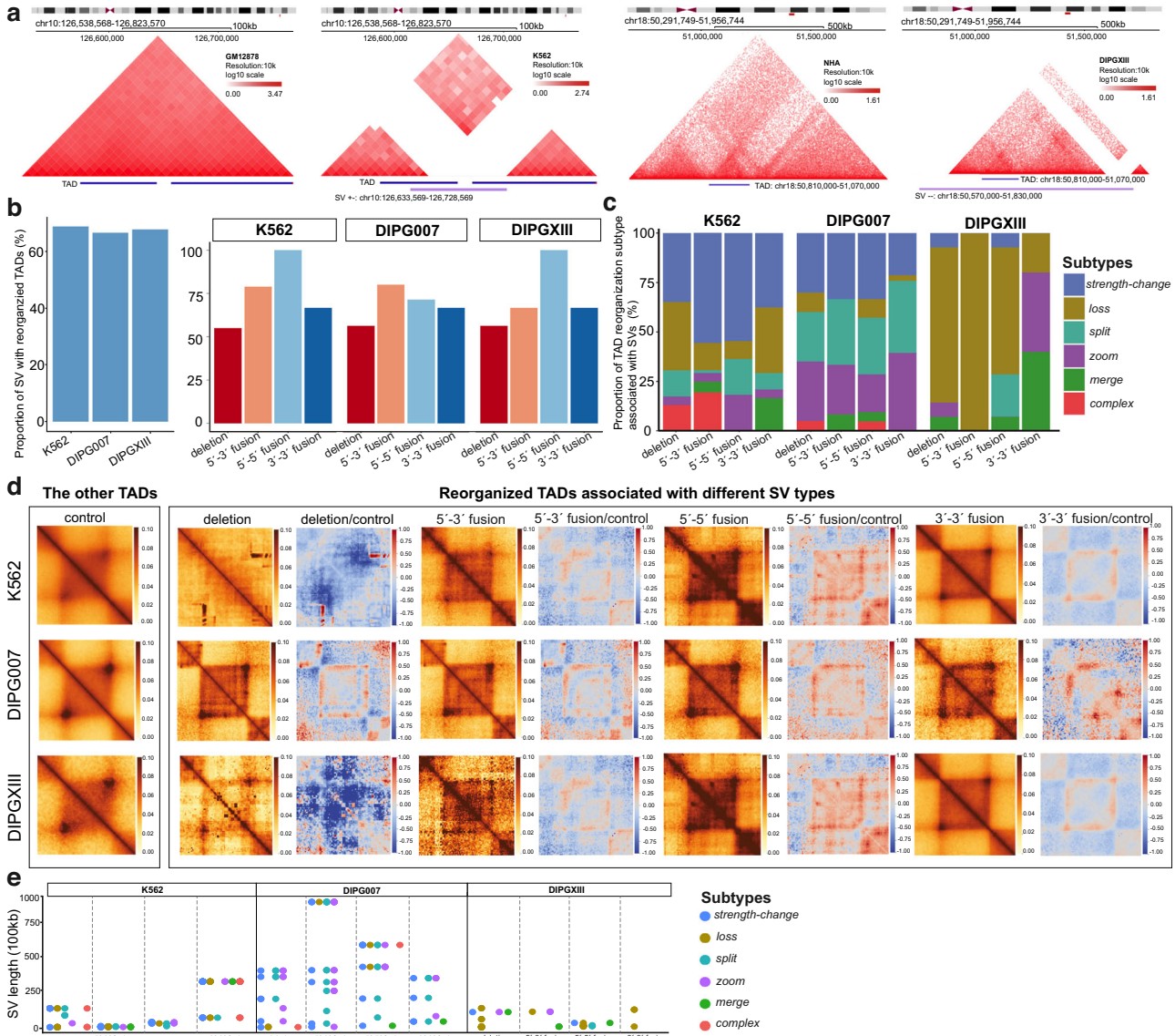

**Fig. 4 | Associations between SVs and reorganized TADs in cell lines with disease. a** Heatmaps showing two example SVs with associated reorganized TADS. *Left* two heatmaps showing the Hi−C contact maps from GM12878 and K562 for a specific K562 SV region, while *Right* two heatmaps showing the Hi−C contact maps from NHA and DIPGXIII for a specific DIPGXIII SV region. The first track below the heatmaps outlines the reorganized TAD regions, and the second track shows the SV regions. **b** Barplot showing the proportions of SVs with associated reorganized TADs across the three cell lines. *Left* barplot includes all SVs; *Right* barplots are stratified by the four types of SV. **c** Stacked barplot showing the proportions of reorganized TAD subtypes associated with each SV type. **d** APA plot summarizing the aggregated changes in reorganized TADs associated with each type of SV. *First column* APA plot summarizing the other TADs (not reorganized) using Hi−C data from condition 2 as a control. *Second column* APA plot summarizing reorganized

TADs associated with deletion SVs using Hi−C data from condition 2. *Third column* Heatmap showcasing log2-transformed fold-change of APA matrices, using APA matrices from the *second column* and the *first column*. Subsequent columns are APA plots summarizing reorganized TADs associated with 5′ to 3′ fusion (*fourth and fifth column*), 5′ to 5′ fusion, and 3′ to 3′ fusion. Rows represent comparisons including GM12878 vs. K562, NHA vs. DIPG007, and NHA vs. DIPGXIII. **e** Jitterplot showing the length of SVs associated with each reorganized TAD subtype across the three cell lines. The APA matrices are based on 25 kb resolution Hi−C data produced by FAN-C using the command 'fanc aggregate -m -p −pixels 90 -r -e −rescale'. The APA plots are generated using python function 'sns.heatmap'. Abbreviations: '+−', deletion; '−+', 5′ to 3′ fusion; '−−', 5′ to 5′ fusion; '++', 3′ to 3′ fusion. NHA, normal human astrocytes; DIPG007 and DIPGXIII, pediatric high-grade glioma cell lines.

significantly enhance our understanding of the relationship between SVs and TADs compared to the previous study[47], further highlighting the biological relevance of reorganized TADs.

### DiffDomain improves profiling of TAD reorganization related to SARS-CoV-2 infection

Severe acute respiratory syndrome coronavirus 2 (SARS-CoV-2) caused over 640 million confirmed coronavirus disease 2019 (COVID-19) cases, including over 6.6 million deaths, worldwide as of December 2, 2022, posing a huge burden to global public health. Wang et al.[48] is the

first Hi−C study into the effects of SARS-CoV-2 infection on host 3D genome organization, finding a global pattern of TAD weakening after SARS-CoV-2 infection. However, the analysis uses aggregation domain analyses which cannot directly identify individual weakened TADs, in addition to missing other subtypes of TAD reorganization.

To further demonstrate the biological applications of DiffDomain, we reanalyze the data. We find that 20.58% (840 in 4082) mock-infected A549-ACE2 TADs are reorganized in SARS-CoV-2-infected A549-ACE2 cells. Among the reorganized TADs, *strength-change* TADs are the leading subtype (64.64%) (Fig. 5a), which is consistent with the

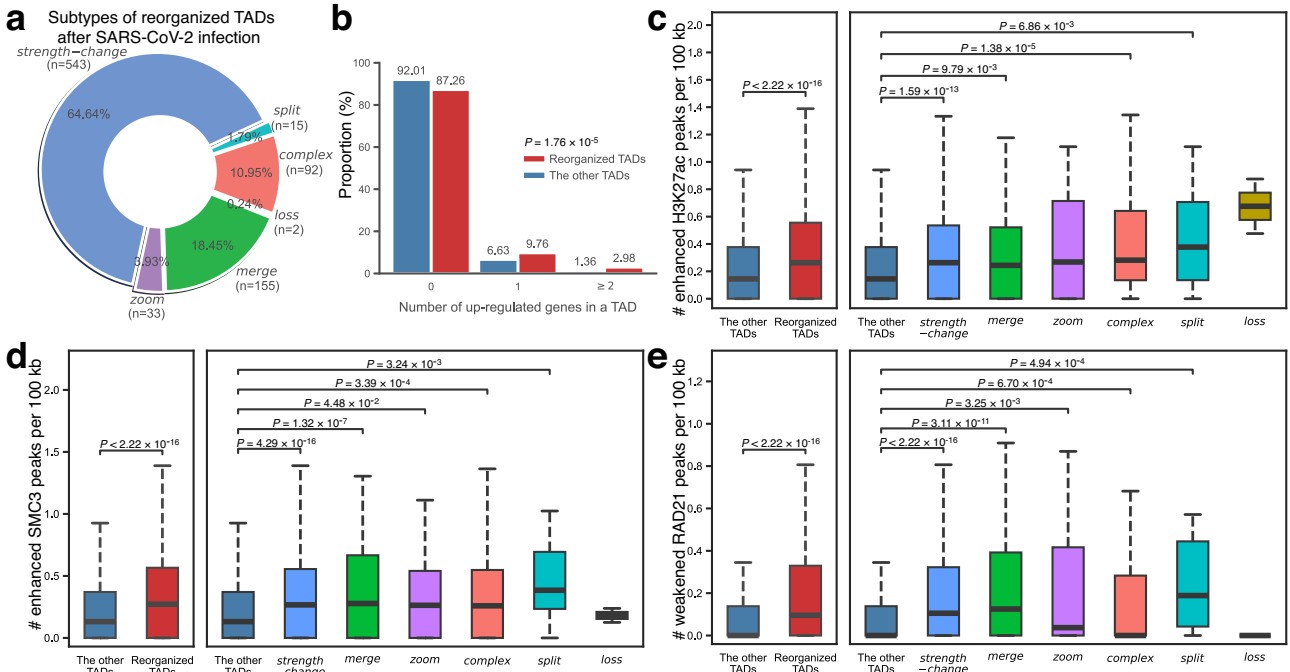

**Fig. 5 | Mock-infected A549-ACE2 TADs that are reorganized in SARS-CoV-2-infected A549-ACE2.** **a** Pie chart showing the percentages of subtypes of reorganized TADs in A549-ACE3 after SARS-CoV-2 infection. *Strength-change* TADs are the leading subtype. **b** Barplot comparing the number of upregulated genes in the reorganized TADs and the other TADs. *x*-axis represents TADs categorized based on the number of upregulated genes located within them, *y*-axis represents the proportion of TADs. Boxplots comparing the numbers of enhanced H3K27ac peaks **c** enhanced SMC3 peaks **d** and weakened RAD21 peaks **e** per 100 kb. *Left*: comparing reorganized TADs with the other TADs (*x*-axis); *right*: comparison stratified

by the subtypes of reorganized TADs (*x*-axis). *y*-Axis represents the number of differential peaks per 100 kb within TADs. *P* values are computed using one-sided Mann–Whitney *U* test, a nonparametric test dealing with asymmetric distributions. The number (*n*) of reorganized TADs in each subtype is presented in (**a**). In the box plots, the middle line represents the median; the lower and upper lines correspond to the first and third quartiles; and the upper and lower whiskers extend to values no farther than $1.5 \times$ IQR. *P* values smaller than $2.22 \times 10^{-16}$ are denoted by $P < 2.22 \times 10^{-16}$.

global pattern of TAD weakening[48], verifying the reorganized TADs identified by DiffDomain. The following most frequent subtypes are *merge* TADs and *complex* TADs (18.45% and 10.95%, Fig. 5a), refining the characterization of TAD reorganization after SARS-CoV-2 infection. These reorganized TADs also enable refined profiling of transcriptional regulation in response to SARS-CoV-2 infection. Compared with the other TADs, the reorganized TADs have significantly higher numbers of upregulated genes and downregulated genes ($P \le 1.27 \times 10^{-4}$, Fig. 5b, Supplementary Fig. 26a). Similar significant patterns are observed comparing *strength-change* TADs and *split* TADs with the other TADs ($P \le 8.07 \times 10^{-3}$, Supplementary Fig. 26b), highlighting that the two subtypes have stronger connections with differentially expressed genes than other subtypes of reorganized TADs. In contrast, compared to the other TADs, the six subtypes of reorganized TADs have significantly higher numbers of both enhanced and weakened peaks of H3K27ac, SMC3, and RAD21 where H3K27ac is a marker for active enhancers and SMC3 and RAD21 are two critical cohesin subunits that regulate 3D genome organization (Supplementary Note 8, Fig. 5c–e, Supplementary Fig. 26c–e). Gene-centric analysis shows that differentially expressed genes in reorganized TADs have stronger connections with differential chromatin interactions than in other TADs. In particular, the *strength-change* subtype has a 3-fold higher proportion (9.73%) of downregulated genes with both enhanced and weakened chromatin interactions compared to other TADs (Supplementary Note 9, Supplementary Fig. 27). These results suggest that, after SARS-CoV-2 infection, the subtypes of reorganized TADs all have strong associations with epigenome reprogram, and *strength-change* TADs and *split* TADs have strong associations with deregulation of gene expression, highlighting the importance of subtypes of reorganized TADs identified by DiffDomain.

## DiffDomain characterizes cell-to-population and cell-to-cell variability of TADs using scHi-C data

Recent advances in scHi-C sequencing methods enable profiling of 3D genome organization in individual cells, revealing intrinsic cell-to-cell variability of TADs among individual cells. However, quantifying the variability is challenging due to the properties of scHi-C data, such as high sparsity, low genome coverage, and heterogeneity[49,50]. As a proof-of-concept, we apply DiffDomain to a moderate-sized scHi-C dataset from mouse neuronal development (median number of contacts per cell at 400,000)[51].

We first ask how many individual cells are sufficient to identify reorganized TADs between cell types with high reproducibility using pseudo-bulk Hi-C data (Supplementary Method 6.1). To do this, we design a sampling experiment to gradually increase the number of used individual cells, and the reproducibility in identified reorganized TADs is quantified using the Jaccard index (Supplementary Method 6.2). We find that DiffDomain can identify reorganized TADs between cell types with reasonable reproducibility (average Jaccard index $\ge 0.104$) using as few as one hundred sampled cells (Fig. 6b, c). For example, DiffDomain consistently identifies that a neuronal TAD, harboring neuronal marker genes *GM24071*, *LRFN2*, *MOCS1*, and *1700008K24RIK*[51], is reorganized in oligodendrocytes with numbers of cells starting at 100 (Fig. 6a). Consistent of DiffDomain on other example genomic regions are shown in Supplementary Fig. 28. On average, DiffDomain identifies that 19.25% neuronal TADs are reorganized in oligodendrocytes using only 250 sampled cells from each cell type, consistent (average Jaccard index at 0.49) with the identified reorganized TADs using all available cells in both cell types (Fig. 6b). Similar results are observed when identifying neonatal neuron 1 (the youngest structure type) TADs that are reorganized in cortical L2–5

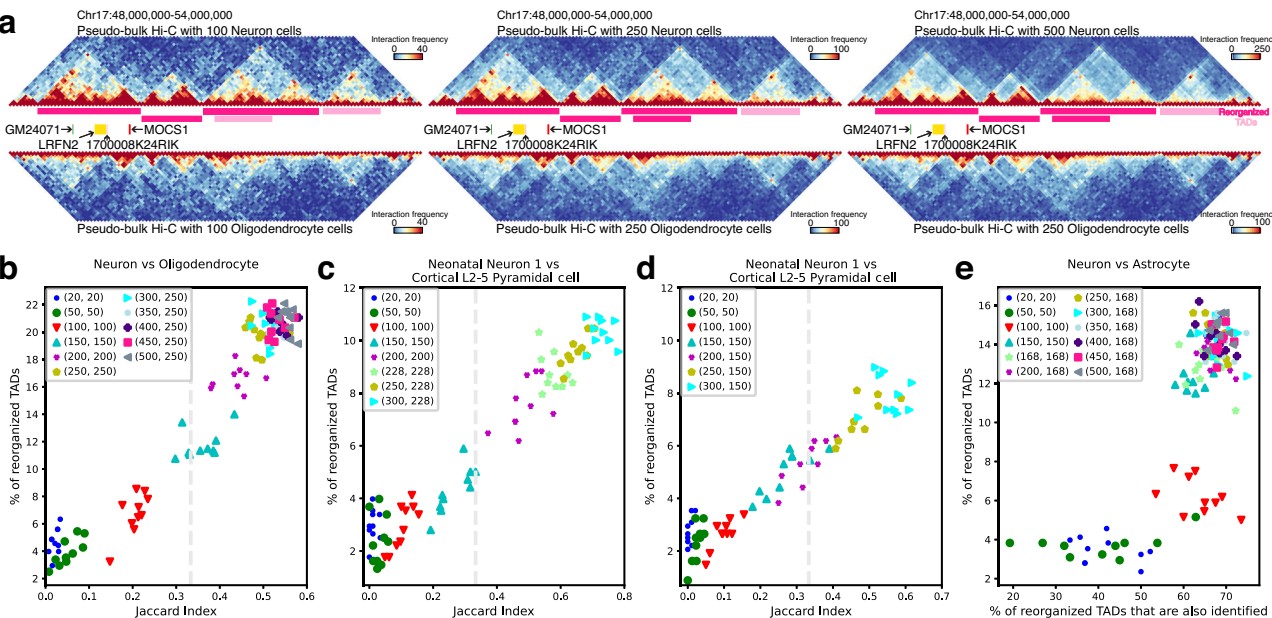

**Fig. 6 | Application of DiffDomain on scHi–C data to identify reorganized TADs between cell types. a** Visualization of the pseudo-bulk Hi–C contact maps and the identified neuronal TADs that are reorganized in oligodendrocytes (dark pink horizontal bars) using varied numbers of randomly sampled individual cells. Gene track shows four neuronal marker genes. **b** Scatter plot showing the proportions of neuronal TADs that are reorganized in oligodendrocytes using varied numbers ($k_1$, $k_2$) of randomly sampled individual cells from the two cell types. Their agreements with the set of reorganized TADs identified using all cells in each cell type are quantified by the Jaccard index ($x$-axis). The vertical dashed line is $JI = 1/3$, representing that two equal-sized sets share half of reorganized TADs. **c, d** Scatter plots showing the proportions of neonatal neuron 1 TADs that are reorganized in cortical L2–5 pyramidal cells. Up to 228 and 150 cortical L2–5 pyramidal cells are randomly sampled, respectively. **e** Scatter plot showing the agreements between sets of reorganized TADs that are identified using (1) pseudo-bulk Hi–C data and (2) bulk Hi–C data.

pyramidal cells (adult type) (Fig. 6c, d). Jointly increasing the numbers of sampled cells in both cell types improves the performance of Diff-Domain, as expected (Fig. 6b). In contrast, only increasing the number of sampled cells in one cell type has a limited boost in performance. For example, oligodendrocytes have only 257 cells, but neurons have 1380 cells. Further increasing the number of sampled neurons from 250 to 500 has a slight performance improvement (Fig. 6b). The observation is further confirmed when comparing neuronal subtypes neonatal neuron 1 and neonatal cortical L2–5 pyramidal cells, in which the number of sampled cells in the latter subtype is no more than 150 (Fig. 6c) or 228 (Fig. 6d). Repeating the analysis by using bulk Hi–C data[52,53] to create gold-standard reorganized TADs, we observed similar patterns in neuronal TADs that are reorganized in astrocytes. For example, sampling 150 cells in each cell type identifies 12.40% neuronal TADs that are reorganized in astrocytes on average. Among the reorganized TADs, 62.55% are also identified as reorganized TADs when bulk Hi–C data are used (Fig. 6e). Considering the median number of contacts per cell at 400000, the merged Hi–C data from hundreds of cells are ultra-sparse pseudo-bulk Hi–C data. These results demonstrate that DiffDomain can work with ultra-sparse Hi–C data.

Next, we move to quantify the cell-to-population variability of TADs, that is, comparing TADs in individual cells to the population average. To do this, scHi–C data with 50 kb resolution from neonatal neuron 1 cells is imputed by scHiCluster[54]. For each TAD, DiffDomain compares the imputed Hi–C contact map of the TAD in each cell to the pseudo-bulk Hi–C contact map. Resulted $P$ values reflect cell-to-population variability of TADs and thus are used by hierarchical clustering to divide TADs into three categories: high, median, and low cell-to-population variational TADs (Fig. 7a). We find that TADs have clear differential cell-to-population variability. One example high cell-to-population variational TAD and its adjacent median cell-to-population variational TADs in 9 cells are shown in Fig. 7c. Among the 2146 neonatal neuron 1 TADs, 8.90% (191) are high cell-to-population variational

TADs, 7.50% (161) and 83.60% (1794) are median and low cell-to-population variational TADs (Fig. 7b). They are distributed across chromosomes (Fig. 7d). Similar results are observed in other cell types (Supplementary Fig. 29). These results demonstrate that TADs have clear differential variability between individual cells and the population average, consistent with earlier observations[26,55].

Next, we move to investigate the cell-to-cell variability of TADs. Requiring only one Hi–C contact matrix from each condition, DiffDomain can directly quantify cell-to-cell variability of TADs between individual cells using imputed scHi–C data. Note that, similar to other methods, pairwise comparison of TADs using scHi–C data from thousands of individual cells leads to exponential growth in runtime and thus is computationally expensive[49]. As a proof of concept, we apply DiffDomain to scHi–C data from randomly selected 50 cortical L2–5 pyramidal cells and 50 adult astrocytes. We find that the cell-to-cell variability of TADs is heterogeneous. The heterogeneity is consistent among the pairwise comparisons but quite different from those from random scenarios in which equal-numbered reorganized TADs are randomly assigned in pairwise comparisons. The proportion of reorganized TADs is consistent among the 2500 pairs of individual cells, ranging from 46.0% to 75.7% (Fig 7e). Across the 50 cortical L2–5 pyramidal cells, the proportions of TADs that are reorganized in a varied number of adult astrocytes are fairly consistent (Fig. 7f). Moreover, the proportions of TADs that are either low in cell-to-cell variability (reorganized in no more than 10 adult astrocytes) or high in cell-to-cell variability (reorganized in more than 40 adult astrocytes) are much higher than those from random scenarios (Fig. 7f, Supplementary Fig. 30), consistent with differential cell-to-population variability of TADs as reported in the previous paragraph. This observation is also in concordance with the randomized placement of TAD-like blocks in individual cells but with a strong preference for TAD boundaries observed in bulk Hi–C data[18], further demonstrating the utilization of DiffDomain.

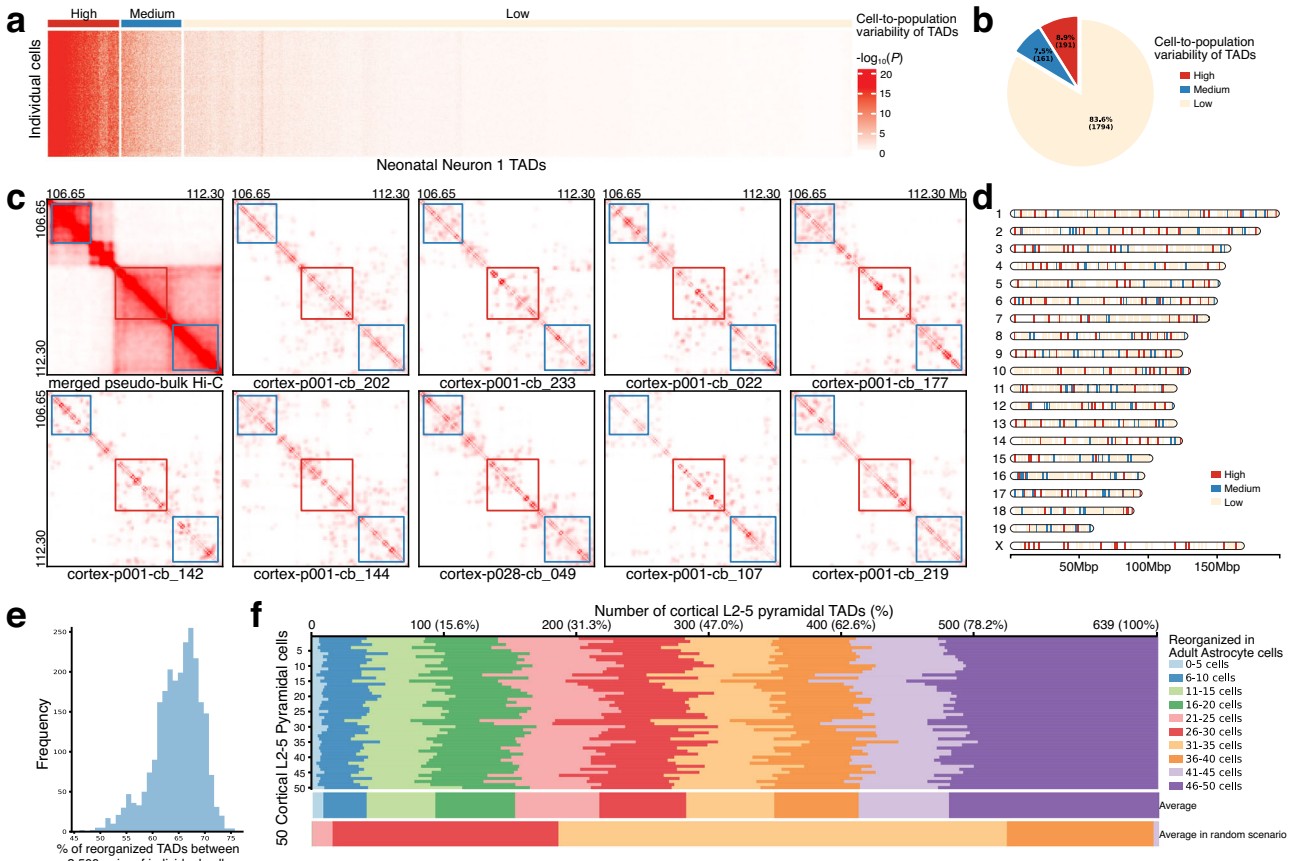

**Fig. 7 | Application of DiffDomain on scHi-C data to characterize cell-to-population and cell-to-cell variability of TADs. a** Heatmap showing high, median, and low cell-to-population variational TADs. *P* value is from comparing scHi-C contact map of a TAD in an individual cell to the pseudo-bulk Hi-C contact map that represents the population average. Classification of TADs is done by hierarchical clustering. **b** Pie chart showing the percentages of the high, median, and low cell-to-population variational TADs. **c** Heatmaps visualizing one high cell-to-population variational TAD (middle red rectangular box) and two median cell-to-population variational TADs (top-left and bottom-right blue rectangular boxes). **d** Chromosome map showing the genomic locations of high, median, and low cell-to-population variational TADs. **e** Histogram showing the percentages of reorganized TADs in 2500 pairwise comparisons of 50 cortical L2–5 pyramidal cells and 50

adult astrocytes. **f** Stacked bar graph showing the number (the percentage) of the cortical L2–5 pyramidal TADs that are reorganized in a varied number (0–50) of adult astrocytes. *y*-Axis represents the selected 50 cortical L2–5 pyramidal cells, indexed from 1 to 50. When comparing a specific cortical L2–5 pyramidal cell, such as cell 1, with 50 adult astrocytes, each cortical L2–5 pyramidal TAD is reorganized in a different number of adult astrocytes. *x*-Axis (top of the plot) is the number (proportion) of the cortical L2–5 pyramidal TADs that are reorganized in 0–5 adult astrocytes, 6–10 adult astrocytes, and subsequent ranges (legend). Averaging the data in the stacked bar graph across the 50 cortical L2–5 pyramidal cells (*y*-axis) results in the first stacked bar graph below, labeled as "Average" on the right. The subsequent stacked bar graph below is the average computed in random scenarios (Supplementary Fig. 30), labeled as "Average in random scenarios" on the right.

In summary, DiffDomain works on scHi-C data to identify reorganized TADs between cell types, identify TADs with differential cell-to-population variability, and characterize cell-to-cell variability of TADs.

## Discussion

In this work, we present a statistical method, DiffDomain, for comparative analysis of TADs using a pair of Hi-C datasets. Extensive evaluation using real Hi-C datasets demonstrates clear advantages of DiffDomain over alternative methods for controlling false positive rates and identifying truly reorganized TADs with much higher accuracy. Applications of DiffDomain to Hi-C datasets from different cell lines and disease states demonstrate that reorganized TADs are enriched with structural variations and associated with CTCF binding site changes and other epigenomic changes, revealing their condition-specific biological relevance. By applying it to a scHi-C dataset from mouse neuronal development, DiffDomain can identify reorganized TADs between cell types with considerable reproducibility using pseudo-bulk Hi-C data from as few as a hundred cells. Moreover, DiffDomain can reliably characterize the cell-to-population and cell-to-cell variability of TADs using scHi-C data.

The major methodological contribution of DiffDomain is directly characterizing the differences between Hi-C contact matrices using high-dimensional random matrix theory. First, DiffDomain makes no explicit assumption on the input chromatin contact matrices, directly applicable to both bulk and single-cell Hi-C data. Second, DiffDomain computes the largest eigenvalue $\lambda_N$ of a properly normalized difference contact matrix $\boldsymbol{D}$, enabling the quantification of the differences of a TAD using all chromatin interactions within the TAD. Third, leveraging the asymptotic distribution of $\lambda_N$, DiffDomain computes theoretical *P* values, which is much faster in computation than simulation methods used in alternative methods. The model assumptions are realistic (Methods). Last but not least, the normalized difference contact matrix $\boldsymbol{D}$ can help pinpoint genomic regions with increased or decreased chromatin interactions within the reorganized TAD, enabling model interpretation and refined integrative analysis with other genomic and epigenomic data.

There is room for improvement. First, DiffDomain has the highest accuracy in detecting truly reorganized TADs, but it misses some truly reorganized TADs that only show subtle structural changes (see example in Fig. 2d, g). Developing more powerful model-based methods is

future work. The manually created list of gold-standard reorganized TADs is deposited in the GitHub repository that hosts the source code, which would benefit the research community for better method development. Second, because of the hierarchy of TADs and sub-TADs[2,33], generalizations of DiffDomain to explicitly consider dependencies among TADs to further refine reorganized TADs identification and classification is future work. Third, it would be desirable to generalize DiffDomain to compare other TAD-like domains[56–61]. scHi-C data is imputed by scHiCluster[54] for the characterization of cell-to-population and cell-to-cell variability of TADs. Benchmarking the effects of different imputation algorithms, including Higashi[26,62] and scVI-3D[63], on quantifying cell-to-population and cell-to-cell variability of TADs is future work.

As a subset of TADs, the identified reorganized TADs could be a critical unit for refined integrative analyses of multi-omics data. We demonstrate this type of application on different human cell lines and disease states, including SARS-CoV-2-infected A549-ACE2 cells. Applying DiffDomain to investigate the connections between TAD reorganization and changes in H3K27me3 modification, a marker recently implicated in development and disease[64,65], is a valuable future work. Notably, future work integrating multiple types of omics data and functional perturbation experiments[66] is necessary to elucidate the causal relationships between TAD reorganization and disease.

DiffDomain is an interpretable statistical method for enhanced comparative analysis of TADs and it works for both bulk and single-cell Hi-C data. The accelerated application of Hi-C and scHi-C mapping technologies would generate ever-growing numbers of bulk and single-cell Hi-C data from different health and disease states. DiffDomain and its future generalizations would be an essential part of the Hi-C analysis toolkit for the emerging comparative analysis of TADs, which in turn would advance understanding of the genome's structure-function relationship in health and disease.

## Methods

In this section, we first introduce the first part of DiffDomain: a model-based method to identify reorganized TADs. We state the model assumptions and their verification using real Hi-C data before reporting the second part of DiffDomain: classification of reorganized TADs into six subtypes. Last is missing value imputation.

### Model-based method to identify reorganized TADs

In this paper, our aim is to identify a subset of TADs that are reorganized between two biological conditions, such as a pair of healthy and diseased cell lines/tissues. Specifically, given a set of TADs identified in one biological condition, we aim to identify the subset of TADs that are reorganized in another biological condition. To achieve this goal, we develop DiffDomain that takes a set of TADs and their Hi-C contact matrices as the input. The TADs are identified using the Arrowhead method[33] (Supplementary Method 1), and Hi-C contact matrices specific to each TAD region are extracted from the genome-wide KR-normalized Hi-C contact maps unless specified otherwise. The core of DiffDomain is converting the comparison of Hi-C contact matrices into a hypothesis-testing problem on their difference matrix. This difference matrix is modeled as a symmetric random matrix, enabling DiffDomain to borrow well-established theoretical results in high-dimensional random matrix theory.

Before explaining the hypothesis testing problem, we first introduce some mathematical notations and normalization operations. For each TAD in biological condition 1, let $N$ denote the number of consecutive and equal-length chromosome bins within the genomic region covered by the TAD. Let $A_1 = (A_{ij}^{(1)}) \in R_{\geq 0}^{N \times N}$ represent the symmetric KR-normalized Hi-C contact matrix, where $A_{ij}^{(1)}$ represents the non-negative Hi-C contact frequency between chromosome bins $i$ and $j$ ($1 \leq i, j \leq N$) in the TAD region in condition 1. In other words, $A_1$ serves as the Hi-C contact matrix specific to the TAD region in biological condition 1, forming a submatrix within the genome-wide Hi-C contact matrix.

Similarly, $A_2 = (A_{ij}^{(2)}) \in R^{N \times N}$ denotes the KR-normalized Hi-C contact matrix corresponding to the same TAD region but in biological condition 2. It is well-known that the Hi-C contact frequency $A_{ij}$ exponentially decreases with an increased linear distance between bins $i$ and $j$. We first log-transform the Hi-C contact matrices $A_1$ and $A_2$ and compute their entry-wise differences, denoted by $D$, as shown in Eq. (2).

$$D = \log(A_1) - \log(A_2). \tag{2}$$

Values in Hi-C contact matrices $A_1$ and $A_2$ could have large differences because of variations in reading depths. For example, the GM12878 Hi-C experiment has 4.76 times more Hi-C contacts than the K562 Hi-C experiment (Supplementary Table 2). Among the 889 GM12878 TADs on Chromosome 1, the averages in the 889 $D$s range from 1.479 to 2.611, with a median at 2.229. To adjust for the differences due to variations in read depths, we normalize $D = (D_{ij})_{i,j=1}^{N}$ by standardizing each of its $k$-off diagonal blocks by

$$(D_{ij} - \hat{\mu}_k)/\hat{\sigma}_k, \tag{3}$$

where $k = j - i$, $2 - N \leq k \leq N - 2$, $\hat{\mu}_k$ and $\hat{\sigma}_k$ are the sample mean and standard deviation of $(D_{mn})_{1 \leq m, n \leq N, n-m=k}$. Here, without abuse of notations, we continue to use $D$ to denote the resulted normalized difference matrix. Note that the normalization is TAD-specific because two different TADs most likely have different $\hat{\mu}_k$ and $\hat{\sigma}_k$, $2 - N \leq k N - 2$. Besides visualization in Fig. 1a–c, the effects of the above procedures for both bulk and single-cell Hi-C matrices from the same TAD are also visualized in Supplementary Fig. 1.

Intuitively, if a TAD does not undergo structural reorganization from biological condition 1 to biological condition 2, the differences between $A_1$ and $A_2$ are caused by multiple factors, including variations in read depths and random perturbations of 3D genome organization. Thus, we assume that entries in $D$ follow a standard Gaussian distribution, resulting in $D$ being a symmetric random noise matrix with entries that follow a standard Gaussian distribution. Scaling $D$ by $\sqrt{N}$, where $N$ represents the number of bins in the TAD, results in $D/\sqrt{N}$ exhibiting characteristics typical of a well-studied random matrix known as a generalized Wigner matrix. With the justifications presented in the next subsection, we assume that $D/\sqrt{N}$ is a generalized Wigner matrix. The problem of identifying reorganized TADs is reformulated as the following hypothesis testing problem:

$H_0 : D/\sqrt{N}$ resembles a generalized Wigner matrix,

$H_1 : D/\sqrt{N}$ does not resemble a generalized Wigner matrix.

The largest eigenvalue of $D/\sqrt{N}$, denoted by $\lambda_N$, converges to 2 with increased $N$[67]. This result helps us to reformulate the hypothesis testing problem as the following:

$$H_0 : \lambda_N = 2 \text{ vs. } H_1 : \lambda_N > 2. \tag{4}$$

Under $H_0$, $\theta_N = N^{2/3}(\lambda_N - 2)$, a normalized $\lambda_N$, converges in distribution to Tracy-Widom distribution with index $\beta = 1$, denoted as $TW_1$[68].

$$\theta_N \xrightarrow{d} TW_1. \tag{5}$$

In other words, under $H_0$, a TAD does not undergo structural reorganization in biological condition 2. Then, the fluctuations of $\theta_N$ is governed by Tracy-Widom distribution $TW_1$. Thus, we choose $\theta_N$ as the test statistic and compute a one-sided $P$ value by

$$P \text{ value} = P_{TW_1}(\theta_N \geq x). \tag{6}$$

A smaller $P$ value means that the TAD is more likely to be reorganized in condition 2.

For a set of TADs, *P* values are adjusted for multiple comparisons by a few methods, with the BH method as the default. The pseudocode of our DiffDomain algorithm is presented in Supplementary Method 2.

## Model assumptions and their verifications

Given two KR-normalized Hi−C contact matrices, DiffDomain computes the normalized difference matrix $D$, bypassing complicated further normalization of individual Hi−C contact matrices[69]. Thus, DiffDomain makes no explicit assumptions on the individual Hi−C contact matrices. DiffDomain only makes assumptions on the normalized difference matrix $D$. First, under $H_0$, DiffDomain assumes that $D/\sqrt{N}$ is a generalized Wigner matrix: a symmetric random matrix with independent mean zero upper diagonal entries. Symmetry is satisfied by $D/\sqrt{N}$ because Hi−C contact matrices are symmetric. The independence assumption on the upper diagonal entries is violated by $D/\sqrt{N}$ considering the well-known fact that Hi−C contact frequencies positively correlate with each other among nearby chromosome bins. However, the violation of the independence assumption does not substantially alter the properties of $D/\sqrt{N}$ and DiffDomain (Supplementary Note 1). Briefly, the empirical properties of $D/\sqrt{N}$ and DiffDomain resemble the following theoretical properties: (i) empirical spectral distribution of a generalized Wigner matrix converging to the well-established semicircle law[70], (ii) $\lambda_N \to 2$[67], (iii) unadjusted *P* values following a uniform distribution when $H_0$ is true and model assumptions are satisfied (Supplementary Note 1, Supplementary Fig. 3). The key result (5) requires one more assumption. It holds under the condition that the distributions of entries in generalized Wigner matrices have vanishing third-moments as *N* tends to infinity[71]. After the standardization procedure (3), $D_{ij}/\sqrt{N}$ approximately follows a Gaussian distribution $N(0, 1/N)$ whose third moment is 0, satisfying the vanishing third-moment assumption. Taken together, assumptions of DiffDomain are appropriate. For additional references on the generalized Wigner matrix, please refer to the comprehensive books authored by Bai and Silverstein[72] and Couillet and Liao[73].

## Reorganized TAD classification

Once a subset of reorganized TADs is identified, the classification of reorganized TADs is critical to interpreting TAD reorganization and linking them to the dynamics of genome functions. Motivated by classifications in previous studies[10,30,31], DiffDomain classifies reorganized TADs into six subtypes: *strength-change*, *loss*, *split*, *merge*, *zoom*, and *complex*. TADs are hierarchically organized, as identified by methods such as Arrowhead. Large TADs can subdivide into smaller TADs, and a genomic region may belong to multiple TADs, complicating reorganized TAD classification. To address this, we compare the TAD list in condition 1 with the TAD list in condition 2, utilizing combinations of identical TADs and overlapping TADs between the two conditions to distinguish the distinct reorganized TAD subtypes (Fig. 1f, Supplementary Method 3). A brief description of the subtypes is provided below.

1. *Strength-change* represents that the boundaries of the reorganized TAD are the same in both conditions. Specifically, the reorganized TAD in condition 1 has a one-to-one identical relationship with a TAD in condition 2.
2. *Loss* represents that the reorganized TAD in condition 1 does not overlap with or be identical to any TADs in condition 2.
3. *Split* represents that a reorganized TAD in condition 1 is split into at least two TADs in biological condition 2. Specifically, the reorganized TAD has either a one-to-many identical relationship or a one-to-many overlapping relationship with TADs in condition 2.
4. *Merge* represents that the reorganized TAD has a many-to-one identical or a many-to-one overlapping relationship with a TAD in condition 2. Specifically, the reorganized TAD and at least one of its adjacent/overlapping TADs in condition 1 are identical to or overlap with a single TAD in condition 2. *Merge* is the opposite of *split* when condition 2 is treated as condition 1.

5. *Zoom* represents that the reorganized TAD in condition 1 has a one-to-one overlapping relationship with a TAD in condition 2, addition to not being identical to any TADs in condition 2.
6. *Complex* represents the other reorganized TADs.

After the classification of reorganized TADs into the six subtypes, the *strength-change* TADs are further subdivided into two categories. Within a *strength-change* TAD, the Hi−C contact frequencies either increase or decrease in biological condition 2, after proper normalization on the differences in total sequenced reads. Subsequently, a *strength-change* TAD can be classified into a *strength-change up* TAD or a *strength-change down* TAD. Before explaining the classification, a few mathematical notions are introduced. Given a *strength-change* TAD, let $m_1$ be the median value of KR-normalized Hi−C contact frequencies within the strength-change TAD in condition 1, $m_2$ be the median value of the KR-normalized Hi−C contact frequencies within the same TAD region but in condition 2. Let $s_1$ be the sum of the KR-normalized Hi−C contact frequencies across all condition 1 TADs, $s_2$ be the sum of the KR-normalized Hi−C contact frequencies across all condition 2 TADs. If the *strength-change* TAD satisfies $\frac{m_1}{m_2} \times \frac{s_2}{s_1} \geq 1$, it is classified as a *strength-change up* TAD. Otherwise, the *strength-change* TAD is classified as a *strength-change down* TAD.

## Missing value imputation

Missing values may exist in Hi−C contact matrices $A_1$ or $A_2$ for a specific TAD region, and their origin can vary. DiffDomain distinguishes between missing values caused by SVs and those caused by other factors, such as low sequencing depth. When SVs are present, say in condition 2, DiffDomain first checks if an $m \times m$ submatrix, $m \geq 3$, of $A_2$ contains exclusively missing values. In such cases, the $m \times m$ submatrix is imputed with a constant, with the default value of 1. Otherwise, if any row and column with a proportion of missing values greater than a given threshold, with a default value of 0.5, DiffDomain removes the corresponding row/column from both $A_1$ and $A_2$. Subsequently, the remaining missing values are imputed by the median contact frequency of interactions at the same distance within the corresponding contact matrix.

## Reporting summary

Further information on research design is available in the Nature Portfolio Reporting Summary linked to this article.

## Data availability

All datasets used in this study are publicly available. Hi−C data of multiple human cell lines and replicates of GM12878 cell line are downloaded from the Gene Expression Omnibus (GEO) database under accession code GSE63525 [https://www.ncbi.nlm.nih.gov/geo/query/acc.cgi?acc=GSE63525][33]. Hi−C data of patient-derived DIPG, NHA, and GBM cell lines and DIPG frozen tissue specimens are downloaded from the GEO database under accession code GSE162976 [https://www.ncbi.nlm.nih.gov/geo/query/acc.cgi?acc=GSE162976][74]. Hi−C data of mock-infected and SARS-CoV-2-infected A549-ACE2 cells are downloaded from the GEO database under accession code GSE179184 [https://www.ncbi.nlm.nih.gov/geo/query/acc.cgi?acc=GSE179184][48]. Processed single-cell Dip-C data of multiple cell types in mouse brains are downloaded from the GEO database under accession code GSE162511 [https://www.ncbi.nlm.nih.gov/geo/query/acc.cgi?acc=GSE162511][51]. The other Hi−C data, DNase-seq data, super-enhancers, and cancer genes are downloaded from the GEO, 4DN, OncoKB, and GeneCards databases, and the data sources are listed in Supplementary Method 5. All relevant analyzed data is available upon request.

## Code availability

The software is published under the GNU GPL v3.0 license. The source code of DiffDomain is available at https://github.com/Tian-Dechao/diffDomain or at this https://doi.org/10.5281/zenodo.10205208[75].

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

## Acknowledgements

This work was supported by the National Natural Science Foundation of China grant 12271536 (D.T. and Z.B.), National Key Research and Development Program of China grant 2021YFC2300102 (D.T.), GuangDong Basic and Applied Basic Research Foundation grant 2022A1515010043 (D.T.), Shenzhen Sustainable Research grant KCXFZ20211020172545006 (D.T.), National Natural Science Foundation of China grant 12171198 (Z.B.), and Jilin Provincial Foundation grant 20210101147JC (Z.B.). We thank Jiang Hu for the helpful discussion on the theoretical properties of the proposed method, Jun Ding and Yang Zhang for the helpful discussion that improved the paper Jian Ma for helpful comments to improve the paper.

## Author contributions

Conceptualization: X. Zhu and D.T.; Methodology: M.G., X. Zhang, and D.T.; Software: M.G., D.H., X. Zhang, and D.T.; Investigation: D.H., M.G., X. Zhang, Y.D., H.X., and D.T.; Writing-Original Draft: D.T.; Writing-Review and Editing: L.Q., X.D., Z.B., X. Zhu, and D.T.; Funding Acquisition: Z.B. and D.T.

## Competing interests

The authors declare no competing interests.
