## [Peer Review File · Nature Communications]

DiffDomain enables identification of structurally reorganized topologically associating domainsREVIEWER COMMENTS

Reviewer #1 (Remarks to the Author):

The authors proposed a statistical method for Domain identification in both bulk and single-cell HiC data. The robustness of this algorithm was reviewed by examining its application on Hi-C data from different cell types and disease states. The authors in their method proposed a two-in-one algorithm where the first is a model-based algorithm to identify reorganized TADs, and the second algorithm introduces a classification of the reorganized TADs into six subtypes. The results and observations from the authors' work are commendable but I have the comments below:

Review Comments:

- Line 420-421, how is the white noise matrix related to the Wigner-noise?
- Line 392: DiffDomain takes a set of TADs and their Hi-C contact matrices as the input to find reorganized TADs. Also, the main manuscript should contain details about the computational algorithms used and the supplementary can provide more details as is currently. The parameter settings used for the TAD algorithms should be mentioned in the supplementary.
- Too many details about the proposed methods are excluded from the manuscript, especially in section B of the supplementary. Some of these details can be mentioned in the manuscript to provide a grounded understanding by reading the paper only.
- In line Line 393 - 395, the authors wrote "The core of DiffDomain is converting the comparison of Hi-C contact matrices into a hypothesis testing problem on their difference matrix, enabling us to borrow well-established theoretical results in high-dimensional random matrix theory" what does the author mean here? It is unclear what the authors mean, especially the part about "borrowing well-established theoretical results in high-dimensional random matrix theory"
- In Lines 396 to 404: There is no mention of if the Hi-C matrices used are normalized (with algorithms such as Yaffe and Tanay, KR, ICE, or SCN) in the methods section. However, in line 442, it is mentioned that the matrices are normalized. For the sake of easier comprehension and answering preprocessing questions earlier on for readers, the authors should state in the first mention in the section "Method-based method to identify reorganized TADs" that the Hi-C IF matrices are normalized. The results section certainly has more details about the algorithm than the methods section.
- Also, is it an assumption that the number of TADs identified between chromosomes from two biological conditions will be the same? As we know it, the number of TAD between two or more independent runs of a TAD algorithm do not report the same number of TADs but with some standard deviation. So, in the context of Line 408, where 889 TADs were reported for both contact matrices A1 and A2 was there a variance, or was some patching done to arrive at the same number? - As a follow-up, how should users process creating the D matrix, if the inputs HiC contact matrices A1 and A2 do not have the same number of detected TADs?

Reviewer #2 (Remarks to the Author):

COMMENT TO AUTHORS

The authors present DiffDomain a novel algorithm for the bioinformatic analysis of Hi-C data to identify differential TADs.

The rationale and implementation of the approach are interesting, but the presentation of results is at times unclear as detailed in specific comments below.

MAJOR POINTS

- the input data to DiffDomain requires a set of called TADs and the authors adopt the Arrowhead algorithm to call TADs. Arrowhead is a very specific TAD calling algorithm that is generally calling few well demarcated TADs. The authors showed data for the comparison of false positive rates when using another TAD calling algorithm (TopDom). In some cases the FPR is higher with Arrowhead TADs (figure S4C). Isn't this something unexpected given the larger number of TAD calls by TopDom? Did you by any chance test also how results would differ on the True Positive Rate estimation on the reference set of rearranged TADs?

- My understanding is that the "truly reorganized TADs" are defined based on literature where different articles adopted different methods to define these differential TADs. We may expect that within this "gold standard" set, the ability to confirm individual TAD changes will actually vary depending on the original way they were defined. Did the author take this into account?

- Did the authors verify with specific analyses if the "reorganized" TAD show in figure 3 is actually the result of a structural variant? In relation to this, in the article section on structural variants, the authors present as a positive result the "enrichment" of structural variants break points in the reorganized TADs. It may be expected that a structural variant is resulting in a change of interactions within a TAD. Therefore, rather than an "enrichment" maybe we should expect that all SV may be associated to a "reorganized" TAD. How many SVs (and what type of SV) do not result in a change of TADs as well?

- line 282: "reasonable reproducibility". I would encourage the authors to quantify this reproducibility in less ambiguous terms.

- in the description of methods, the authors explicitly state that the approach is based on the assumption that the matrix "D resembles a white noise matrix, and that D/\sqrt{N} is a generalized wigner matrix". However, in the paragraph with additional details, the authors explicitly state that the assumption is not completely valid as there is a "violation of the independence assumption" (line 450). Some additional analyses to this concerns are presented in figure S2 but they are not clearly discussed in the main text.

- For all the "reorganized" TADs there's usually only one example visualized but sometimes it is difficult to see an evident change in a single TAD heatmap. It would be nice to show a sort of average TAD change (metaprofile) summarizing the change across many TADs of the same category (e.g. "strength-change" category)

- In figure legends there are usually several missing details, as not all of the figure elements are explained. For example in figure 3, there are several rectangular elements, supposedly marking the regions of changes, but they are not clearly explained. Like wise there are several genomic tracks not explained in the legend. Just as an example, figure 3C panel is just described as "The normalized difference matrix D between the two cell lines highlights the differences in Hi-C contact maps." without other explanations. In general all the figure legends lack details that should be explicitly mentioned to avoid ambiguities. For example, in figure 4, there are several green bars, that maybe are marking structural variants (or maybe not, as they are never explicitly mentioned in the legend). If they mark structural variants, it is not clear what type of SV (amplification, deletion, inversion, translocation?)

- also related to this, the plots of some specific TADs in relation to structural variants in figure 4 is a bit anecdotal. A sort of summary (e.g. metaprofile) of changes in TADs in relation to a set of SVs of the same type would be more informative.

- Figure S12 is very complex and based on the figure legend I could not understand it. All the elements in the figure should be more clearly explained in the legend.

- in figure S14 and other related analyses, the "changes" in TADs are compared with changes in chromatin accessibility, but this analysis seems to put all "reorganized" TADs together, i.e. not discriminating between TADs with increased or decreased contacts and/or accessibility, that may have different correspondence on the changes in accessibility.

- in figure S19 and other related analyses, the reorganization of TADs is correlated to the presence of Structural Variants, but this is done without making distinctions between different SVs and how they are expected to change the TADs profile. Throughout the results it would be important to account for the different expected patterns in relation to SVs and TADs reorganization as there are also different classes of reorganized TADs.

- likewise, in figure S20 there's a barplot showing downregulated genes in reorganized TADs. It would be important to understand how interactions change (up or down regulated).

- the colorbar gradients in several figures do not have a label to state what are the numeric values: e.g. see figures S21 and S22

- I can't understand the figure legend of figure S23. The legend is mentioning multiple stacked bars, but it is not always clear to which one is referred each sentence. Also the orange gradient in the larger heatmap (? or stacked barplot?) is not clearly marked: what are the different shades of color?

MINOR POINTS

- The algorithm defines different classes of reorganized TADs: e.g. "merge", "split", "zoom" etc. The explanation for these different labels is a bit hidden in the methods. It would be useful to give at least a brief explanation of the meaning of these labels in the main text.

- line 197 "VEFGA"  "VEGFA"

- in figure S13 samples are clustered based on the "number of split TADs". Is this based just on numbers, and not on their actual location (i.e. could it be that different TADs are split and counted as if they coincide)? Why not using just a continuous quantitative score (e.g. insulation score) to cluster samples in this figure?

- figure S22 legend: "top left chromosome map"  maybe it was "bottom right chromosome map"? Likewise the heatmaps in the figure do not have a label to explicitly state what is in the columns.

MS #: NCOMMS-22-52455-T

MS TITLE: DiffDomain enables identification of structurally reorganized topologically associating domains

MS AUTHORS: Dunming Hua, Ming Gu, Xiao Zhang, Yanyi Du, Li Qi, Xiangjun Du, Zhidong Bai, Xiaopeng Zhu and Dechao Tian

Statement of the revisions

We appreciate the comments and suggestions from all reviewers. In response, we have made revisions to the manuscript (see our point-by-point response below). Major updates based on the reviewers' suggestions include:

1. We have conducted a comprehensive investigation into the associations between SVs and reorganized TADs. This has unveiled that the associations between SVs and TAD reorganization are specific to SV types and diseases, enhancing our understanding of the relationship between SVs and TADs.
2. We have added method comparison based on the original way that the truly reorganized TADs are defined, further highlighting the advantage of DiffDomain over alternative methods.
3. We have added aggregated peak analyses (APA) and summary plots to provide a comprehensive visual representation of TAD reorganization within the same subtypes and in relation to different SV types, facilitating interpretation and understanding of the results.
4. We have added new results demonstrating distinct associations of TAD reorganization subtypes with epigenomic changes, further highlighting the importance of TAD reorganization subtypes in understanding genome activity and functionality.
5. We have clarified and expanded explanations in data preprocessing, method assumptions, method clarification, results description, and figure explanation to enhance the clarity, comprehensibility, and accuracy of the paper.

Overall, these revisions collectively elevate the quality of the paper by clarifying concepts, providing more comprehensive analyses, and making the research more accessible and relevant to a wider audience. We believe our work has been significantly strengthened.

Point-by-point response to comments from the reviewers

Reviewer #1

The authors proposed a statistical method for Domain identification in both bulk and single-cell HiC data. The robustness of this algorithm was reviewed by examining its application on Hi-C data from different cell types and disease states. The authors in their method proposed a two-in-one algorithm where the first is a model-based algorithm to identify reorganized TADs, and the second algorithm introduces a classification of the reorganized TADs into six subtypes. The results and observations from the authors work are commendable but I have the comments below:

Response: We thank the reviewer for pointing out that our method is robust, mathematical equations are reasonable and explainable, and workflow is clear.

Line 420-421, how is the white noise matrix related to the Wigner-noise?

Response: Thank you for pointing this out. The statement omits necessary details including entries in D following standard Gaussian distribution and D being symmetric. To enhance clarity, the necessary

details are included in revised the manuscript. Please refer to the revision on page 11, lines 464-469 in the revised main text.

Line 392: DiffDomain takes a set of TADs and their Hi-C contact matrices as the input to find reorganized TADs. Also, the main manuscript should contain details about the computational algorithms used and the supplementary can provide more details as is currently. The parameter settings used for the TAD algorithms should be mentioned in the supplementary.

Response: We thank the reviewer for this constructive comment. The method used for calling TADs, namely Arrowhead, along with the normalization method KR for Hi-C contact matrices have been included right after Line 392 for enhanced clarity. Please refer to the revision on page 10, lines 431-433 in the revised main text.

To provide comprehensive details about the TAD algorithms, the algorithms and their parameter settings are now provided in the Supplementary Methods A.4. Please refer to the revisions on page 3, lines 920-931 in the Supplementary File.

Too many details about the proposed methods are excluded from the manuscript, especially in section B of the supplementary. Some of these details can be mentioned in the manuscript to provide a grounded understanding by reading the paper only.

Response: We fully agree with the reviewer's comment regarding the necessity for further elucidation in the main text. To improve clarity, we expanded upon the explanations for Supplementary Methods and Supplementary Results within the main text. The revisions include

- Moved the Reorganized TAD classification (Supplemental Methods A.2) to the Methods section in the main text. Please refer to the revision on page 12, lines 512-530.
- Expanded the explanations to the statement corresponding to Supplementary Results B.1. Please refer to the revision on page 3, lines 115-119.
- Expanded the explanations to the statement corresponding to Supplementary Results B.4. Please refer to the revision on page 6, lines 222-225.
- Expanded the explanations to the statement corresponding to Supplementary Results B.5. Please refer to the revision on page 6, lines 226-228.
- Expanded the explanations to the statement corresponding to Supplementary Results B.7. Please refer to the revision on page 6, lines 239-242.

In line 393 - 395, the authors wrote “The core of DiffDomain is converting the comparison of Hi-C contact matrices into a hypothesis testing problem on their difference matrix, enabling us to borrow well-established theoretical results in high-dimensional random matrix theory” what does the author mean here? It is unclear what the authors mean, especially the part about “borrowing well-established theoretical results in high-dimensional random matrix theory”

Response: Thank you for pointing this out. We added the sentence, “This difference matrix is modeled as a symmetric random matrix”, immediately after mentioning of the difference matrix. This revision provides a smoother transition from the difference matrix to its connection with random matrix theory. Please refer to the revision on page 10, lines 435-436 in the revised main text.

In Lines 396 to 404: There is no mention of if the Hi-C matrices used are normalized (with algorithms such as Yaffe and Tanay, KR, ICE, or SCN) in the methods section. However, in line 442, it is mentioned that the matrices are normalized. For the sake of easier comprehension and answering preprocessing questions earlier on for readers, the authors should state in the first mention in the section “Method-based method to identify reorganized TADs” that the Hi-C IF matrices are normalized. The results

section certainly has more details about the algorithm than the methods section.

Response: We fully agree with this constructive comment. The input Hi-C data is KR normalized. The information is added after the first mention in the section “Method-based method to identify reorganized TADs” as suggested, please refer to the revision on page 10, line 433, line 440, and line 444 in the revised main text.

Also, is it an assumption that the number of TADs identified between chromosomes from two biological conditions will be the same? As we know it, the number of TAD between two or more independent runs of a TAD algorithm do not report the same number of TADs but with some standard deviation. So, in the context of Line 408, where 889 TADs were reported for both contact matrices A_1 and A_2 was there a variance, or was some patching done to arrive at the same number? - As a follow-up, how should users process creating the D matrix, if the inputs HiC contact matrices A_1 and A_2 do not have the same number of detected TADs?

Response: We deeply apologize for the confusion caused by our previous writing. DiffDomain does not assume that the number of TADs identified between chromosomes from two biological conditions will be the same. We fully acknowledge the reviewer’s point that independent runs could result in different numbers of called TADs, and this variability becomes even more pronounced when comparing different biological conditions. This variability was a driving factor that motivated the design of DiffDomain.

In DiffDomain, the comparison is not predicated on the exact matching of the number of TADs between conditions. Rather, DiffDomain focuses on testing whether a TAD identified in biological condition 1 is significantly reorganized in another biological condition. This is practical. A TAD identified in one condition could undergo multiple subtypes of reorganization in another condition, such as splitting into several smaller TADs or merging with adjacent TADs to form a larger TAD (see Fig. 1F for visualization of different subtypes of reorganizations).

To provide further clarify, the Hi-C contact matrices A_1 and A_2 used in DiffDomain are not genome-wide Hi-C contact matrices, but rather submatrices specific to individual TAD regions. Specifically, for each TAD identified in condition 1, let N denote the number of consecutive and equal-length chromosome bins within the genomic region covered by the TAD. $A_1 = (A_{ij}^{(1)}) \in R_{\geq 0}^{N \times N}$ represents the KR-normalized Hi-C contact matrix, where $A_{ij}^{(1)}$ represents the non-negative Hi-C contact frequency between chromosome bins i and j ($1 \leq i, j \leq N$) within the TAD region of condition 1. In other words, A_1 serves as the Hi-C contact matrix specific to the TAD region in biological condition 1, forming a submatrix within the genome-wide Hi-C contact matrix. Similarly, $A_2 = (A_{ij}^{(2)}) \in R_{\geq 0}^{N \times N}$ denotes the KR-normalized Hi-C contact matrix corresponding to the same TAD region, but in biological condition 2. Several paired A_1 s and A_2 s corresponding to a few TADs are visualized in Fig. 1F and Fig. 2D. This design ensures that each TAD in biological condition 1 is independently analyzed without assuming a predefined match in TADs between biological conditions.

It is crucial to clarify that A_1 and A_2 are not genome-wide or chromosome-level Hi-C contact matrices which contain multiple TADs. Rather, A_1 and A_2 are TAD-specific Hi-C contact matrices, with each TAD corresponding to a unique pair of A_1 and A_2 matrices. As for the context of the mentioned 889 TADs, they are the TADs identified on chromosome 1 in GM12878 cell line (condition 1). For each of these 899 TADs, DiffDomain extracts the specific Hi-C contact matrix A_1 from GM12878, and the corresponding Hi-C contact matrix A_2 from K562, both tailored to the same TAD region. In total, the 899 TADs yield 899 A_1 submatrices extracted from GM12878 and 899 A_2 submatrices from K562.

In response to the follow-up question on D matrix, please note that A_1 and A_2 are the Hi-C contact matrices specific to the same TAD region but from different conditions. A_1 and A_2 have the same

dimensions, both being $N \times N$ for a TAD with N bins. This guarantees that the difference matrix D can always be calculated.

We have taken steps to clarify these aspects. Please refer to the revision in the overview of DiffDomain on page 3, lines 86-89 and the revised explanations on page 11, lines 438-445 in the Methods section in the revised main text.

We sincerely apologize for the confusion and greatly appreciate the valuable feedback provided by the reviewer once again.

Reviewer #2

The authors present DiffDomain a novel algorithm for the bioinformatic analysis of Hi-C data to identify differential TADs. The rationale and implementation of the approach are interesting, but the presentation of results is at times unclear as detailed in specific comments below.

MAJOR POINTS

The input data to DiffDomain requires a set of called TADs and the authors adopt the Arrowhead algorithm to call TADs. Arrowhead is a very specific TAD calling algorithm that is generally calling few well demarcated TADs. The authors showed data for the comparison of false positive rates when using another TAD calling algorithm (TopDom). In some cases the FPR is higher with Arrowhead TADs (figure S4C). Isn't this something unexpected given the larger number of TAD calls by TopDom? Did you by any chance test also how results would differ on the True Positive Rate estimation on the reference set of rearranged TADs?

Response: Regarding higher false positive rate (FPR) using Arrowhead TADs than TopDom TADs, the observation is to some extent anticipated. The observation occurs when considering Hi-C replicates “GM12878_primary” and “GM12878_replicate” as condition 1, where their TADs are tested whether they are reorganized in other conditions. Closer examination reveals that both “GM12878_primary” and “GM12878_replicate” have more Arrowhead TADs than TopDom TADs (figure S4A). Specifically, “GM12878_primary” and “GM12878_replicate” have 8270 and 8049 Arrowhead TADs, respectively, in contrast to 7456 and 7393 TopDom TADs. Given the higher number of Arrowhead TADs in both replicates, the higher FPR observed with Arrowhead TADs is to some extent anticipated results. We have included a discussion on this observation in the revised Supplementary Results B.2 on page 6, lines 1044-1047 of the Supplementary File. Additionally, we have addressed a typographical error in Figure S4A, where one of the X-axis tick labels “GM12878_replicate” has been accurately changed to “GM12878_combined”. The figure S4 is now referenced as Supplementary Fig. S5 in the updated Supplementary File.

Regarding the estimation of the True Positive Rate using a different reference set of reorganized TADs, we fully understand the significance of this comment but we have not collected a second reference set. The primary rationale behind this choice is that the current reference set of reorganized TADs is sufficient. The set contains 65 truly reorganized TADs that are compared across 146 pairwise comparisons, providing comprehensive representations on levels of TAD reorganization, with 26 pairs involving domain-level changes, 77 pairs involving boundary-level changes, and 43 pairs involving loop-level changes. Additionally, the set provides comprehensive representations of Hi-C files ($n = 57$) and publications ($n = 15$). Combined with additional results listed down below, we believe that it adequately facilitates a faithful estimation of True Positive Rate, ensuring a fair method comparison.

Regarding the variation in the True Positive Rate, indeed, variation is observed across the three groups

of reorganized TADs. Specifically, the True Positive Rate of DiffDomain is the highest (84.6%) on TADs with domain-level changes, modest (74.0%) on TADs with boundary-level changes, and the lowest (48.8%) on TADs with loop-level changes. This variation in the True Positive Rate is also observed in alternative methods. The variation is expected. Across the three groups of truly reorganized TADs, the TADs have increased similarities between their corresponding Hi-C contact maps between biological conditions. Importantly, when compared with alternative methods, DiffDomain consistently outperforms them by achieving the highest True Positive Rate across the three groups of truly reorganized TADs. In contrast, the second-best method varies among the three groups. The classification of truly reorganized TADs into the groups of domain-level change, boundary-level change, and loop-level change is inspired by the subsequent comment. For related revisions and detailed information, please refer to our response to the next comment.

My understanding is that the “truly reorganized TADs” are defined based on literature where different articles adopted different methods to define these differential TADs. We may expect that within this “gold standard” set, the ability to confirm individual TAD changes will actually vary depending on the original way they were defined. Did the author take this into account?

Response: We greatly thank the reviewer for this insightful comment. Following the suggestion, based on the original way the truly reorganized TADs were defined, they are broadly categorized into three distinct groups: domain-level change, boundary-level change, and loop-level change. Briefly, domain-level change represents extensive alternations in interactions within a TAD, boundary-level change indicates changes in TAD boundaries, and loop-level change represents alternations in loops within a TAD. These groups have increased similarities in Hi-C contact maps, as demonstrated by increased SCC scores between biological conditions. Indeed, the TRP of DiffDomain decreases across the three groups. However, DiffDomain consistently achieves the highest TPR, while the second-best method varies, demonstrating the advantages of DiffDomain over alternative methods. We have incorporated this analysis in the revised manuscript, specifically on page 5, lines 174-180 and page 5, line 186 of the revised main text. Additionally, further details can be found on page 2, lines 893-905 and Supplementary Fig. S11 in the revised Supplementary File. The full list of truly reorganized TADs with their group information is uploaded to Github at https://github.com/Tian-Dechao/diffDomain/blob/main/data/gold_standard_reorganized_TADs.tsv and provided as a new Supplementary Table, Supplementary Table S6, as an Excel.

Did the authors verify with specific analyses if the “reorganized” TAD show in figure 3 is actually the result of a structural variant? In relation to this, in the article section on structural variants, the authors present as a positive result the “enrichment” of structural variants break points in the reorganized TADs. It may be expected that a structural variant is resulting in a change of interactions within a TAD. Therefore, rather than an “enrichment” maybe we should expect that all SV may be associated to a “reorganized” TAD. How many SVs (and what type of SV) do not result in a change of TADs as well?

Response: Regarding the reorganized TAD in Fig 3, indeed, the reorganized TAD lies in an SV region. The SV is a long-range duplication (chr6:16.77-51.73 Mb) covering an approximately region of 34.97 Mb. The corresponding Hi-C contact map features increased interactions around the two breakpoints of the SV, highlighted with a black box in Fig. R1. In contrast, the TAD (chr6:43.61-44.05 Mb) covers a much smaller region which spanning around 0.44 Mb. Together, it seems unlikely that the SV contributes to the reorganization of this particular TAD.

Regarding the connection between reorganized TADs and SVs, we totally agree with the reviewer. Following the suggestions provided in this and other related comments, we conducted an in-depth analysis and used the Nucleome Browser to visualize each SV along with their associated TADs. This visual examination revealed a bug in DiffDomain. Specifically, DiffDomain does not distinguish between missing

Figure R1: Visualization of SV and reorganized TAD reported in Fig 3 in the main text. **(A)** Heatmap visualizing the Hi-C contact map in GM12878. **(B)** Heatmap visualizing the Hi-C contact map in K562. The first track below the heatmaps denotes the TAD that are reorganized in K562. The second track below the heatmap in **(B)** represents the SV region. SV label ‘++’ represents 3’ to 3’ fusion, a type of duplication. The rectangle boxes in the heatmaps highlight the major difference between the two Hi-C contact maps that are associated with the SV.

values caused by SVs such as deletions and those caused by other factors such as low sequencing depth. One unique feature of a deletion SV in Hi-C contact map is that a submatrix of the Hi-C contact map is missing. An example is visualized in the revised Fig. 4A. We fixed this bug. Now, in situations where a submatrix within a TAD’s Hi-C contact map contains only missing values, a specialized imputation is performed. The missing values in the submatrix are set to a constant, with the default value of 1. This addresses the previous issue where DiffDomain either directly outputs a P value equaling missing value if the missing submatrix involves a large portion of TAD bins or inappropriate imputes the missing values with the median contact frequency of interactions at the same distance. We added the imputation step to the Methods section (page 13, lines 532-540) in the main text and updated the DiffDomain code on Github accordingly.

Following the implementation of the bug-fixing, DiffDomain identified additional 4 SVs in K562, 4 SVs in DIPG007, and 14 SVs in DIPGXIII that are associated with reorganized TADs. However, DiffDomain still identified 8 SVs in K562, 10 SVs in DIPG007, and 4 SVs in DIPGXIII cell lines that remain unassociated with TAD reorganization. Upon individual visualization, we confirmed that among the 8 SVs in K562, 3 SVs have reorganized TADs that are not corrected identified by DiffDomain. This suggests that that the remaining 5 SVs may lack associated TAD reorganization, Similarly, 3 out of the 10 SVs in DIPG007, and 1 out of the 4 SVs in DIPGXIII cell lines may lack associated TAD reorganization. We are unable to provide a definitive explanation for the absence of associated reorganized TADs in these 9 SVs. Nevertheless, we have included the visualizations of these cases in the revised Supplementary File as Supplementary Fig. S23-S25.

The results from enrichment analysis have been updated after the bug-fixing and have been combined together into the Supplementary Fig. S22 in the revised Supplementary File.

Building upon the insights provided by this comment and related comments, we have comprehensively revised the results section on the association between SVs and reorganized TADs. Please refer to page 6 in the main text for the revised results. We thank the reviewer again!

Line 282: “reasonable reproducibility”. I would encourage the authors to quantify this reproducibility in less ambiguous terms.

Response: We thank the reviewer for pointing this out. To clarify the statement, we added the average Jaccard index (≥ 0.104) when 100 cells are sampled in each cell type. Please refer to page 8, line 317 and 320 in the revised manuscript for the revision.

In the description of methods, the authors explicitly state that the approach is based on the assumption that the matrix “ D resembles a white noise matrix, and that D/\sqrt{N} is a generalized wigner matrix”. However, in the paragraph with additional details, the authors explicitly state that the assumption is not completely valid as there is a “violation of the independence assumption” (line 450). Some additional analyses to this concerns are presented in figure S2 but they are not clearly discussed in the main text.

Response: We thank the reviewer for pointing out this. We incorporated the discussion of this violation of independent assumption in the main text. Please refer to the revision on page 3, lines 115-119.

For all the “reorganized” TADs there’s usually only one example visualized but sometimes it is difficult to see an evident change in a single TAD heatmap. It would be nice to show a sort of average TAD change (metaprofile) summarizing the change across many TADs of the same category (e.g. “strength-change” category).

Response: We greatly appreciate the constructive suggestion from the reviewer. In response, we use the aggregation peak analyses (APA) to comprehensively summarize the changes observed across the reorganized TADs of the same subtype in the context of 6 pairs of cell lines. Specifically, The APA plots show consistent changes occurring within the reorganized TADs with the same subtype, which aligns with the defined characteristics of each subtype of TAD reorganization. For example, APA plot for *strength-change* TADs prominently show changes within the aggregated TAD region, supporting the definition of *strength-change* TADs. APA plot for *merge* TADs show increased chromatin interactions between the aggregated TAD region and its adjacent regions, aligning with the *merge* TAD definition.

We added the analysis to the revised manuscript. Please refer to page 3, lines 110-111 in the main text and Supplementary Fig. S2 in the Supplementary File. We thank the reviewer for this constructive suggestion again.

In figure legends there are usually several missing details, as not all of the figure elements are explained. For example in figure 3, there are several rectangular elements, supposedly marking the regions of changes, but they are not clearly explained. Like wise there are several genomic tracks not explained in the legend. Just as an example, figure 3C panel is just described as “The normalized difference matrix D between the two cell lines highlights the differences in Hi-C contact maps.” without other explanations. In general all the figure legends lack details that should be explicitly mentioned to avoid ambiguities. For example, in figure 4, there are several green bars, that maybe are marking structural variants (or maybe not, as they are never explicitly mentioned in the legend). If they mark structural variants, it is not clear what type of SV (amplification, deletion, inversion, translocation?)

Response: We greatly appreciate the reviewer’s feedback. To enhance clarity, we have added more explanations to the captions of Fig. 1-6. For details, please refer to the revisions in captions of Fig. 1-6 in the revised main text.

Also related to this, the plots of some specific TADs in relation to structural variants in figure 4 is a bit anecdotal. A sort of summary (e.g. metaprofile) of changes in TADs in relation to a set of SVs of the same type would be more informative.

Response: We thank the reviewer for the visualization comment. We applied APA plots to visualize aggregated changes in reorganized TADs that are associated with each SV types, which greatly improve the quality of the analysis. For more details, kindly refer to the revised results section on page 6 of the

main text and the corresponding APA plots in the revised Fig. 4.

Figure S12 is very complex and based on the figure legend I could not understand it. All the elements in the figure should be more clearly explained in the legend.

Response: To enhance clarity, we have rearranged the layout of Supplementary Fig. S12 and expanded explanations in the figure caption. For details, please refer to the updated Supplementary Fig. S14 in the revised Supplementary File.

In figure S14 and other related analyses, the “changes” in TADs are compared with changes in chromatin accessibility, but this analysis seems to put all “reorganized” TADs together, i.e. not discriminating between TADs with increased or decreased contacts and/or accessibility, that may have different correspondence on the changes in accessibility.

Response: We thank the reviewer for this constructive feedback. In response to this, we investigated the associations between TAD reorganization subtypes and chromatin accessibility as well as histone modifications. We found that distinct associations between TAD reorganization subtypes and chromatin accessibility as well as histone modifications. Specifically, TAD reorganization subtypes *strength-change up*, *zoom*, *split*, and *complex* are associated with increased chromatin accessibility and histone modifications signals marking active transcription activities. Conversely, TAD reorganization subtypes *loss*, *strength-change down* and *merge* are associated with decreased histone modifications signals marking active transcription activities, emphasizing the importance of TAD reorganization subtypes in investigating genome activity and functionality. For more details, please refer to the revisions on page 6, lines 232-238 in the updated main text, and Supplementary Results B.6 on page 8, lines 1138-1151 and Supplementary Fig. S18 in the updated Supplementary File.

We thank the reviewer again for this constructive comment.

In figure S19 and other related analyses, the reorganization of TADs is correlated to the presence of Structural Variants, but this is done without making distinctions between different SVs and how they are expected to change the TADs profile. Throughout the results it would be important to account for the different expected patterns in relation to SVs and TADs reorganization as there are also different classes of reorganized TADs.

Response: We thank the reviewer again for this insightful comment. Following the suggestion, we conducted additional analysis and have two new findings: 1) each type of SVs has a distinct association with subtypes of TAD reorganization; 2) the association between SVs and TAD reorganization is disease-specific. For more details, please refer to the revised results section on page 6 and the APA plots summarizing aggregated changes in TAD reorganization subtypes across the SV types in the revised Fig. 4 in the main text.

Likewise, in figure S20 there’s a barplot showing downregulated genes in reorganized TADs. It would be important to understand how interactions change (up or down regulated).

Response: We thank the reviewer for this insightful comment. To explore the relationship between down-regulated genes and their Hi-C interactions, we utilized HiC-DC+ to identify differential chromatin interactions. A gene is classified as having a differential chromatin interaction if its transcription start site lies within 50 kb of a chromosome bin involved in such an interaction. Our findings reveal that 36.9% (62 out of 162) of down-regulated genes located in reorganized TADs have at least one differential chromatin interaction, a proportion 1.74 times higher than that of down-regulated genes (21.2%, 92 out of 433) located in the other TADs. Due to the limited number of differential chromatin interactions in down-regulated genes, the down-regulated genes are categorized into four groups based on interac-

tion changes. Notably, down-regulated genes within *strength-change* TADs show a three-fold higher occurrence (9.73%) of down-regulated genes with both enhanced and weakened chromatin interactions compared to other TADs. Distinct associations with differential chromatin interactions are observed for down-regulated genes across the subtypes of reorganized TADs and the other TADs. Similar results are observed when analyzing up-regulated genes, highlighting the role of TAD reorganization in gene expression regulation.

This analysis is added in the revised manuscript. Please refer to page 7, lines 300-304 in the main text, and Supplementary Results B.9 on page 10 and Supplementary Fig. S27 in the Supplementary File.

The colorbar gradients in several figures do not have a label to state what are the numeric values: e.g. see figures S21 and S22.

Response: We thank the reviewer for this comment. In response, we have included the missing labels in the revised manuscript. Specifically, we have added the label “Interaction frequency” to the colorbar gradients in Fig. 6A and Supplementary Fig. S21, as well as the label “ $-\log_{10}(P)$ ” to the colorbar gradients in Supplementary Fig. S22. Please refer to the updated figures in Fig. 6A, Supplementary Fig. S28 and S29 in the revised manuscript.

I can’t understand the figure legend of figure S23. The legend is mentioning multiple stacked bars, but it is not always clear to which one is referred each sentence. Also the orange gradient in the larger heatmap (? or stacked barplot?) is not clearly marked: what are the different shades of color?

Response: We apologize for the confusion caused. To enhance clarity, we have added labels (*A*), (*B*), and (*C*) to the figure, relocated the legend to the bottom of the figure, and expanded explanations in the figure caption. For details, please refer to the updated Supplementary Fig. S30 in the revised Supplementary File.

MINOR POINTS

The algorithm defines different classes of reorganized TADs: e.g. “merge”, “split”, “zoom” etc. The explanation for these different labels is a bit hidden in the methods. It would be useful to give at least a brief explanation of the meaning of these labels in the main text.

Response: We thank this valuable comment. In accordance to the suggestion and other related suggestions, we moved the Reorganized TAD classification (Supplemental Methods A.2) to the Methods section in the main text. Please refer to page 12, lines 512-530 for the revision.

Line 197 “VEFGA” → “VEGFA”

Response: Done.

In figure S13 samples are clustered based on the “number of split TADs”. Is this based just on numbers, and not on their actual location (i.e. could it be that different TADs are split and counted as if they coincide)? Why not using just a continuous quantitative score (e.g. insulation score) to cluster samples in this figure?

Response: We totally agree that the exact location of reorganized TADs or insulation score would achieve fine-scale sample clustering. However, in this analysis, the primary objective is not to cluster samples but to demonstrate the similarity between cell types based on TAD reorganization patterns. The use of the number of *split* TADs for clustering adequately serves our specific objective in this context.

Figure S22 legend: “top left chromosome map” → maybe it was “bottom right chromosome map”? Likewise the heatmaps in the figure do not have a label to explicitly state what is in the columns.

Response: *Top left* is revised to *Bottom right*. Columns in the heatmaps are individual cells. We have added the corresponding cell names in the heatmaps and expanded the explanations of the heatmaps in the figure caption. Additionally, we have added the label “ $-\log_{10}(P)$ ” to the colorbar gradients in Supplementary Fig. S22. Please refer to the updated Supplementary Fig. S29 in the revised Supplementary File.

REVIEWER COMMENTS

Reviewer #1 (Remarks to the Author):

The authors have adequately addressed my comments, and I do not have any additional questions for them.

Reviewer #3 (Remarks to the Author):

The novel algorithm developed by the authors in this study, DiffDomain, is a relevant and useful tool to call differential TADs among conditions. The authors demonstrated the robustness, accuracy and superior performance over the previously published methodology with an extensive set of analyses and applications to publicly available datasets.

In this revised manuscript, the authors successfully addressed the concerns that were raised. However, a few points need to be further clarified and amended.

Main points:

Concerning the Methods section 'Reorganized TAD classification', the brief summary provided is useful to have an intuition of each reorganized TAD subtype. However, additional explanations (possibly in the Supplementary Methods) are needed to understand the exact algorithm and parameters used to quantitatively assign TADs to each class:

- For the strength-change subtype: it is unclear whether, in the formula, the 'medium' value (should it rather be 'median'?) m and the sum of the elements s always refer to the TAD only, or to the entire Hi-C matrix. I imagine that one of the two factors (either the median ratio, or the sum ratio) should be computed on the entire matrix and the other only within the TAD (to achieve the stated 'proper normalization of the differences in total sequenced reads')?

- Additionally, it is stated that if that product is ≥ 1 , the TAD is strength-change up, otherwise it is strength-change down. Then, according to this definition, wouldn't every TAD be classified as strength-change (either up or down) and none as lost/split/merge/zoom/complex? what is the rationale and the parameters to classify as strength-change versus the other categories?

- How is it quantitatively computed that a TAD is 'lost'?

- How are split/merge/zoom subtypes identified? does it require calling TADs on condition 2?

Minor points:

- line 177: SCC acronym is used before its definition (in line 184)
- line 240: “share a substantial” maybe the word ‘fraction’ (or ‘percentage’) is missing
- lines 257-264: it is unclear what are the four SV types (deletions? duplications?) that are mentioned here
- line 532: “missing values may be exist”, ‘be’ should be removed
- line 1119 (methods): specify which statistical test has been used to obtain this mentioned p-value
- Methods: include references to articles treating Wigner matrix theory for readers to further explore

Suggestion for future work:

While the histone modification data analysed in this manuscript by the authors is already sufficient to prove the association of reorganized TADs with epigenetic changes, as a future perspective I recommend looking into TAD reorganization and the role of H3K27me3, since this histone mark and the 3D genome structure have been recently implicated in development and disease (see e.g. Kraft et al. PNAS 2022, Zhenhai Du et al. Molecular Cell 2020)

MS #: NCOMMS-22-52455A

MS TITLE: DiffDomain enables identification of structurally reorganized topologically associating domains

MS AUTHORS: Dunming Hua, Ming Gu, Xiao Zhang, Yanyi Du, Li Qi, Xiangjun Du, Zhidong Bai, Xiaopeng Zhu and Dechao Tian

Statement of the revisions

We appreciate the comments and suggestions from reviewer #3. In response, we have made revisions to the manuscript (see our point-by-point response below). Major updates based on the reviewer's suggestions include:

1. We have added rationales and expanded the description of the classification of reorganized TAD into six subtypes in the Methods section, and have added Supplementary Methods A.2 to provide more detailed explanations. These explanations would further clarify the classification method for identifying reorganized TAD subtypes.
2. We have added the connection between TAD reorganization and the role of H3K27me3 as part of future work in the revised Discussion section.
3. We have addressed all the minor comments raised by reviewer #3.

Overall, these revisions collectively elevate the quality of the paper by clarifying the classification of reorganized TADs and making the research more accessible and relevant to a wider audience.

Point-by-point response to comments from the reviewers

Reviewer #1

The authors have adequately addressed my comments, and I do not have any additional questions for them.

Response: We would like to express our gratitude for taking the time to review our manuscript. We are pleased to hear that you found our responses to your previous comments satisfactory and that you do not have any additional questions. We greatly appreciate your time and expertise in reviewing our work, which has been invaluable in improving the quality of our work.

Reviewer #3

The novel algorithm developed by the authors in this study, DiffDomain, is a relevant and useful tool to call differential TADs among conditions. The authors demonstrated the robustness, accuracy and superior performance over the previously published methodology with an extensive set of analyses and applications to publicly available datasets.

In this revised manuscript, the authors successfully addressed the concerns that were raised. However, a few points need to be further clarified and amended.

Main points:

Concerning the Methods section Reorganized TAD classification, the brief summary provided is useful to have an intuition of each reorganized TAD subtype. However, additional explanations (possibly in the Supplementary Methods) are needed to understand the exact algorithm and parameters used to quantitatively assign TADs to each class:

Response: We thank the reviewer for this valuable suggestion. To enhance the clarity of the classification method, we have expanded the summary within the Methods section. Furthermore, we have provided detailed explanations of reorganized TAD subtypes including the algorithm and parameters used in the Supplementary Methods A.2. For more details, please refer to the revised Methods section (page 12, line 517 to page 13, line 549) in the main text and Supplementary Methods A.2 (page 1, line 878 to page 2, line 934) in the supplementary file.

For the strength-change subtype: it is unclear whether, in the formula, the medium value (should it rather be median?) m and the sum of the elements s always refer to the TAD only, or to the entire Hi-C matrix. I imagine that one of the two factors (either the median ratio, or the sum ratio) should be computed on the entire matrix and the other only within the TAD (to achieve the stated proper normalization of the differences in total sequenced reads)?

Response: We thank the reviewer for pointing this out. Regarding the product of two ratios, the median ratio is based on the median contact frequencies within the TAD region in conditions 1 and 2. The sum ratio is the ratio of the sum of contact frequencies across all TADs identified in condition 2 to the sum across all TADs identified in condition 1. To enhance clarity, we have elaborated on the explanation of the product (page 13, lines 538-548). Additionally, the typo “medium” has been corrected in the revised manuscript.

Additionally, it is stated that if that product is ≥ 1 , the TAD is strength-change up, otherwise it is strength-change down. Then, according to this definition, wouldn't every TAD be classified as strength-change (either up or down) and none as lost/split/merge/zoom/complex? what is the rationale and the parameters to classify as strength-change versus the other categories?

Response: We thank the reviewer for this valuable feedback. First, we would like to highlight that the classification procedure has three consecutive steps: (1) identifying the subset of condition 1 TADs that are reorganized in condition 2; (2) classifying the reorganized TADs into six subtypes including *strength-change* TADs; (3) subdividing only the *strength-change* TADs into either *strength-change up* TADs or *strength-change down* TADs. Following these three consecutive steps, the scenario where every TAD would be classified as *strength-change* (either up or down) would not happen. To better reflect the three consecutive steps and enhance clarity, we have relocated the step (3) as a standalone paragraph right after step (2) in the revised Methods section (page 13, lines 538-549).

The rationale behind classifying reorganized TADs as *strength-change* subtype versus the other subtypes in step (2) is primarily motivated by the hierarchical nature of TADs such as a genomic region can be assigned to multiple TADs in one condition. The hierarchical nature complicates the classification. To address this, we compare the TAD list in condition 1 with the TAD list in condition 2, utilizing combinations of identical TADs and overlapping TADs between the two conditions to distinguish the distinct reorganized TAD subtypes. In broad terms, a *strength-change* TAD represents that its genomic region is also covered by a single TAD in condition 2. Specifically, the *strength-change* TAD has a one-to-one identical relationship with a single TAD in condition 2. Otherwise, any reorganized TAD not satisfying this criterion is classified as one of the other subtypes, depending on its relationship with TADs in condition 2. These information have been included in the revised Methods section (page 12, lines 517-525) and Supplementary Methods A.2 (page 1, lines 878-934).

How is it quantitatively computed that a TAD is lost?

Response: Specifically, if a reorganized TAD does not overlap with or being identical to any TADs in condition 2, it is classified as a *loss* TAD. That is, the genomic region covered by the reorganized TAD do not intersect with genomic regions covered by any TADs in condition 2. We have expanded

the explanation of how *loss* TADs are identified in the Methods section (page 13, lines 526-527) and Supplementary Methods A.2 (page 2, lines 909-910).

How are split/merge/zoom subtypes identified? does it require calling TADs on condition 2?

Response: The identification of *split/merge/zoom* subtypes also depends on the relationship of reorganized TADs and TADs in condition 2. Briefly, if a reorganized TAD has either a one-to-many identical relationship or a one-to-many overlapping relationship with TADs in condition 2, the reorganized TAD is identified as a *split* TAD. Conversely, if a reorganized TAD and at least one of its adjacent/overlapping TADs in condition 1 are identical to or overlap with a single TAD in condition 2, the reorganized TAD is identified as a *merge* TAD. A reorganized TAD is identified as a *zoom* TAD if the reorganized TAD has a one-to-one overlapping relationship with a single TAD in condition 2, additional to not being identical to any TADs in condition 2. To enhance the clarity of the classification method, we have added the above description in the revised Methods section (page 13, lines 528-536) in the main text. Detailed explanations have been included in the Supplementary Methods A.2 (page 1, lines 878-934) in the revised supplementary file.

Indeed, TADs in condition 2 are required by the identification of reorganized TAD subtypes. We have explicitly stated this requirement for the classification of reorganized TADs in the revised Methods section (page 12, lines 519-522) in the main text and in the Supplementary Methods A.2 (page 1, lines 878-880) in the revised supplementary file.

Minor points

line 177: SCC acronym is used before its definition (in line 184)

Response: Thank you for point out this. The acronym SCC is now defined before its first use. For the revision, Please refer to the revision on page 5, lines 177-178, and 184 in the revised main text.

line 240: share a substantial maybe the word fraction (or percentage) is missing

Response: The word “proportion” is added right after the word “substantial” (page 6, line 240).

lines 257-264: it is unclear what are the four SV types (deletions? duplications?) that are mentioned here

Response: We apologize for not specifying the four SV types in our previous version. We have provided explicit clarification by including the exact names of the four SV types before reporting their associations with subtypes of reorganized TADs. For detailed information, please refer to page 7, lines 257-260 in the revised main text.

line 532: missing values may be exist, be should be removed

Response: Done (page 13, line 551).

line 1119 (methods): specify which statistical test has been used to obtain this mentioned p-value

Response: The used hypergeometric test has been added alongside the p-value. For the revision, please refer to page 9, line 1215 in the revised supplementary file.

Methods: include references to articles treating Wigner matrix theory for readers to further explore

Response: We appreciate this valuable suggestion. We have added references [68, 69, 71, 73, 74] to the key theoretical results on Wigner matrix in the revised manuscript. For the references, please refer to page 11, lines 478, 482, 503, 510-512.

Suggestion for future work:

While the histone modification data analysed in this manuscript by the authors is already sufficient to

prove the association of reorganized TADs with epigenetic changes, as a future perspective I recommend looking into TAD reorganization and the role of H3K27me3, since this histone mark and the 3D genome structure have been recently implicated in development and disease (see e.g. Kraft et al. PNAS 2022, Zhenhai Du et al. Molecular Cell 2020)

Response: We thank the reviewer for the valuable literature and future work. We have added them as part of future work in the Discussion section (page 10, lines 414-416) in the revised main text.